# Effects of short-term unloading and active recovery on human motor unit properties, neuromuscular junction transmission and transcriptomic profile

Fabio Sarto[1] , Daniel W. Stashuk[2] , Martino V. Franchi[1,3] , Elena Monti[1] , Sandra Zampieri[1,3,4] , Giacomo Valli[1], Giuseppe Sirago[1] , Julián Candia[5] , Lisa M. Hartnell[5], Matteo Paganini[1], Jamie S. McPhee[6], Giuseppe De Vito[1,3], Luigi Ferrucci[5], Carlo Reggiani[1,7] and Marco V. Narici[1,3,7]

[1]*Department of Biomedical Sciences, University of Padova, Padova, Italy*
[2]*Department of Systems Design Engineering, University of Waterloo, Ontario, Canada*
[3]*CIR-MYO Myology Center, University of Padova, Padova, Italy*
[4]*Department of Surgery, Oncology, and Gastroenterology, University of Padova, Padova, Italy*
[5]*Longitudinal Studies Section, Translational Gerontology Branch, National Institute of Aging, National Institutes of Health, Baltimore, MD, USA*
[6]*Department of Sport and Exercise Sciences, Manchester Metropolitan University Institute of Sport, Manchester, UK*
[7]*Science and Research Center Koper, Institute for Kinesiology Research, Koper, Slovenia*

Handling Editors: Scott Powers & Bruno Grassi

Linked articles: This article is highlighted in a Perspective article by Soendenbroe. To read this article, visit https://doi.org/10.1113/JP283800.

The peer review history is available in the Supporting Information section of this article (https://doi.org/10.1113/JP283381#support-information-section).

*The Journal of Physiology*

**Abstract** Electrophysiological alterations of the neuromuscular junction (NMJ) and motor unit potential (MUP) with unloading are poorly studied. We aimed to investigate these aspects and the underlying molecular mechanisms with short-term unloading and active recovery (AR). Eleven healthy males underwent a 10-day unilateral lower limb suspension (ULLS) period, followed by

21-day AR based on resistance exercise. Quadriceps femoris (QF) cross-sectional area (CSA) and isometric maximum voluntary contraction (MVC) were evaluated. Intramuscular electromyographic recordings were obtained during 10% and 25% MVC isometric contractions from the vastus lateralis (VL). Biomarkers of NMJ molecular instability (serum c-terminal agrin fragment, CAF), axonal damage (neurofilament light chain) and denervation status were assessed from blood samples and VL biopsies. NMJ and ion channel transcriptomic profiles were investigated by RNA-sequencing. QF CSA and MVC decreased with ULLS. Increased CAF and altered NMJ transcriptome with unloading suggested the emergence of NMJ molecular instability, which was not associated with impaired NMJ transmission stability. Instead, increased MUP complexity and decreased motor unit firing rates were found after ULLS. Downregulation of ion channel gene expression was found together with increased neurofilament light chain concentration and partial denervation. The AR period restored most of these neuromuscular alterations. In conclusion, the human NMJ is destabilized at the molecular level but shows functional resilience to a 10-day unloading period at least at relatively low contraction intensities. However, MUP properties are altered by ULLS, possibly due to alterations in ion channel dynamics and initial axonal damage and denervation. These changes are fully reversed by 21 days of AR.

(Received 27 May 2022; accepted after revision 26 August 2022; first published online 7 September 2022)

**Corresponding author** F. Sarto: Department of Biomedical Sciences, University of Padova, Padova, 35131, Italy. Email: fabio.sarto.2@phd.unipd.it.

M. V. Narici: Department of Biomedical Sciences, CIR-MYO Myology Centre, University of Padova, Padova 35131, Italy. Email: marco.narici@unipd.it

**Abstract figure legend** Eleven young males took part in a 10-day unilateral lower limb suspension (ULLS) intervention. This unloading period was followed by 21 days of subsequent active recovery (AR) based on resistance exercise. At baseline, after ULLS and after AR we evaluated muscle size by ultrasound and *in vivo* muscle function by isometric dynamometry. Motor unit potential (MUP) properties and neuromuscular junction (NMJ) transmission stability were assessed using intramuscular electromyography. Finally, vastus lateralis muscle biopsies and blood samples were collected. The ULLS intervention resulted in increased NMJ molecular instability in absence of NMJ transmission stability impairment. Changes in MUP properties were observed, including increased MUP complexity and decreased motor unit firing rates, possibly due to initial axonal damage, partial denervation and altered ion channels dynamics. The AR period was effective in restoring these neuromuscular changes. LS0: baseline data collection; LS10: unilateral lower limb suspension day 10; AR21: active recovery day 21.

## Key points

- We used integrative electrophysiological and molecular approaches to comprehensively investigate changes in neuromuscular integrity and function after a 10-day unilateral lower limb suspension (ULLS), followed by 21 days of active recovery in young healthy men, with a particular focus on neuromuscular junction (NMJ) and motor unit potential (MUP) properties alterations.

- After 10-day ULLS, we found significant NMJ molecular alterations in the absence of NMJ transmission stability impairment. These findings suggest that the human NMJ is functionally resilient against insults and stresses induced by short-term disuse at least at relatively low contraction intensities, at which low-threshold, slow-type motor units are recruited.

- Intramuscular electromyography analysis revealed that unloading caused increased MUP complexity and decreased motor unit firing rates, and these alterations could be related to the observed changes in skeletal muscle ion channel pool and initial and partial signs of fibre denervation and axonal damage.

- The active recovery period restored these neuromuscular changes.

## Introduction

The neuromuscular junction (NMJ) is a highly specialized synapse in the peripheral nervous system enabling electrical transmission between a motor neuron terminal and its postsynaptic skeletal muscle fibre. NMJ maintenance and dynamic remodelling are essentially shaped by neuromuscular activity (Bloch-Gallego, 2015; Wilson & Deschenes, 2005). Physical exercise is well known to maintain NMJ integrity at both its pre- and its postsynaptic components, via the action of different neurotrophins, as reported mostly in animal works (Nishimune et al., 2014). Conversely, seminal studies conducted in animal models have shown NMJ alterations in response to both total disuse (induced by denervation or synaptic blockade via toxin application) (Brown & Ironton, 1977; Eldridge et al., 1981; Labovitz et al., 1984; Pestronk & Drachman, 1978; Pestronk et al., 1976) and partial disuse (muscle unloading, such as hindlimb suspension) (Deschenes & Wilson, 2003; Fahim, 1989; Fahim & Robbins, 1986; Pachter & Eberstein, 1984). In particular, previous work reported changes in endplate size (Deschenes & Wilson, 2003; Eldridge et al., 1981; Labovitz et al., 1984; Pestronk & Drachman, 1978), post-synaptic fold structure (Fahim, 1989; Labovitz et al., 1984), subcellular active zones (Deschenes et al., 2021), acetylcholine (ACh) receptor distribution (i.e. increased expression of extrasynaptic ACh receptors) and density (Eldridge et al., 1981; Pestronk & Drachman, 1978; Pestronk et al., 1976), and nerve terminal sprouting (Eldridge et al., 1981; Fahim & Robbins, 1986; Pestronk & Drachman, 1978; Pestronk et al., 1976). While it is still unclear whether some of these NMJ alterations are reflective of signs of impairment or compensatory mechanisms (Slater, 2020), it is evident that animal NMJs exhibit remarkable remodelling in response to chronic disuse or decreased neuromuscular activity (Wilson & Deschenes, 2005). However, direct translation of these findings obtained in murine models to humans is complex. NMJ structure varies according to phylum and species, and thus animal and human NMJs have different morphology (Hughes et al., 2006; Wood & Slater, 2001). For instance, human NMJs are generally smaller and more 'fragmented' than those in mice (Jones et al., 2017; Slater, 1992) and recent evidence revealed the existence of specific cellular and molecular features in human NMJs (Jones et al., 2017). In addition, the presence of NMJ structures in muscle biopsies obtained with conventional sampling procedures in humans is generally low (Aubertin-Leheudre et al., 2019), making the direct observation of their morphology challenging, particularly for longitudinal studies. For this reason, human NMJ stability is generally assessed indirectly by employing biomarkers of muscle denervation and/or systemic biomarkers of NMJ health (Soendenbroe et al., 2021).

It has been recently suggested that human NMJs may also be affected by disuse/unloading (Monti et al., 2021). An early and partial sign of fibre denervation was reported by two studies, after 3 days of dry immersion (an extreme model of unloading) in healthy males (Demangel et al., 2017) and 14 days of bed rest in middle-aged men and women (Arentson-Lantz et al., 2016). Moreover, our research group recently observed greater NMJ molecular instability (inferred from increased serum C-terminal agrin fragment concentration; CAF) and changes in the expression of some selected genes involved in the NMJ regulation after 10 days of bed rest (Monti et al., 2021). These findings seem particularly relevant from a physiological and clinical perspective as NMJ alterations, together with an altered intracellular calcium handling, may represent an early determinant of force loss during periods of unloading, commonly experienced after injury, surgery and illness (Monti et al., 2021). While there is convincing evidence that NMJs are affected by disuse/unloading from a morphological and molecular perspective, whether NMJ transmission stability (i.e. NMJ function) becomes impaired is, at present, poorly under-stood. In this scenario, intramuscular electromyography (iEMG), and more specifically 'near-fibre (NF) electro-myography', represents a unique opportunity to study *in vivo* human NMJ transmission stability, together with an evaluation of motor unit potential (MUP) properties (size, spatial distribution) and motor unit (MU) firing rates (Piasecki et al., 2021; Sanders et al., 2019).

Hence, we aimed to investigate the neuromuscular changes in response to a short period of unloading

**Fabio Sarto** obtained his Bachelor's and Master's degree in Sports Science at the University of Padova (Italy). During his Master's, he was a visiting student at KU Leuven (Belgium). He is currently a PhD student in Biomedical Sciences at the University of Padova under the supervision of Professor Marco Narici. His main research interest focuses on neuromuscular plasticity in response to physical inactivity, ageing and exercise using an integrative physiological approach.

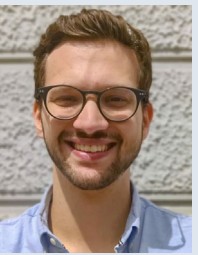

(employing the unilateral lower limb suspension (ULLS) model) and active recovery (AR) in humans with a particular focus on NMJ alterations and MUP characteristics. Our hypotheses were that (i) 10 days of ULLS, would be sufficient to cause NMJ molecular alterations, possibly resulting in NMJ transmission stability impairment, and increased MUP complexity and decreased MU firing rates, in association with loss of muscle function and (ii) 21 days of AR would counteract these neuromuscular changes.

## Methods

### Ethical approval

The present study was conducted in accordance with the standards set by the latest revision of the *Declaration of Helsinki*, and was approved by the Ethics Committee of the Department of Biomedical Sciences of the University of Padova (Italy) (reference number HEC-DSB/01-18). Participants were informed about all the experimental procedures through an interview and an information sheet. Volunteers signed a written consent form and were allowed to drop out of the study at any stage.

### Participants and experimental protocol

Twelve recreationally active young adults (age: 22.1 (2.9) years; height: 1.78 (0.03) m; body mass (72.1) kg) took part in this study. We decided to recruit only male individuals because ULLS can be associated with an increased risk of deep venous thrombosis (Bleeker et al., 2004) which is generally more common in young females. The sample size was defined using *a priori* power analysis calculations (see the 'Statistical analysis' section for further details). Inclusion criteria were 18 to 35 years of age, body mass index between 20 and 28 kg m$^{-2}$ and involvement in recreational physical activities (1–3 times/week). Exclusion criteria were sedentary and very active (>3 training/week) individuals, smokers, history of deep venous thrombosis and acute or chronic musculoskeletal, metabolic and cardiovascular disorders. The protocol included 1 day of familiarization with the experimental procedure and measurements at baseline (LS0), after 10 days of ULLS (LS10) and 21 days of AR based on resistance exercise (AR21). The duration of the AR phase was based on previous observations that full recovery of muscle function after a 2-week lower limb immobilization required a retraining period lasting twice as long (4 weeks) as the unloading phase (Suetta et al., 2009). An intermediate measurement of maximum voluntary contraction (MVC) was performed after 10 days of AR (AR10) to verify how much force had been recovered with respect to baseline values.

Final measurements were performed 3 days after the last training session of the AR phase to avoid potential muscle fatigue. Participants were asked to maintain their habitual diet period throughout the entire intervention period and refrain from coffee and alcohol intake and any form of exercise during the 24 h preceding the data collection at each time point. Each participant performed all the tests at the same time of the day to minimize possible influences of circadian variations.

### Unilateral lower limb suspension

The ULLS model, originally described by Berg et al. (1991), was applied for 10 days. The non-dominant leg of the participants was fitted with a shoe having an elevated sole (50 mm), while the dominant leg was suspended and kept at a slightly flexed position (∼15/20 degrees of knee flexion) using straps. Volunteers were asked to walk with crutches during the whole ULLS period and to refrain from loading the suspended leg in any way. A familiarization session, in which the participants practiced carrying out daily tasks while performing ULLS (Tesch et al., 2016), was completed. Participants were recommended to wear elastic compression socks on the suspended leg during ULLS and to perform passive, range of motion, non-weight-bearing exercises of the ankle, as precautionary measures to prevent deep venous thrombosis (Bleeker et al., 2004). Moreover, an ultrasound-Doppler examination was performed after 5 days of ULLS. Compliance of the participants was evaluated through daily calls and messages and by comparing calf temperature and circumference after 5 and 10 days of ULLS, as previously suggested (Tesch et al., 2016).

### Active recovery

After the ULLS period, participants took part in a 21-day AR programme based on resistance exercise. The training programme started 3 or 4 days after the LS10 measurements in order to grant a period of recovery from the damage induced by the muscle biopsy sampling. Participants trained 3 times per week with training sessions separated by at least 24 h. The AR programme consisted of 10 repetitions of unilateral leg press and leg extension exercises for three sets at 70% of one-repetition maximum (1RM). The 1RM was estimated indirectly since the participants were not previously involved in resistance exercise and underwent 10 days of unloading. Briefly, 4−6RM (i.e. the heaviest load that they could lift and lower under control in the range between 4 and 6 repetitions) was assessed and the 1RM was subsequently estimated using a previously proposed formula (Brzycki, 1993), verifying that the resulting value was

within the range indicated by the National Strength and Conditioning Association (NSCA) training load chart (Beachle & Earle, 1994). The 1RM was reassessed at the first training session of each week and the load employed during the training was adjusted accordingly. Both exercises were executed from full knee extension (0 degrees) to ∼90 degrees limb flexion. Sets were separated by a 2-min rest. The time under tension was set at ∼2 s in both the concentric and the eccentric phases.

### *In vivo* muscle structure and function

**Muscle size measurements.** Muscle cross-sectional area (CSA) of the quadriceps femoris was evaluated using extended-field-of-view ultrasonography imaging (Mylab70, Esaote, Genoa, Italy). A 47 mm, 7.5 MHz linear array transducer was used to collect images at different muscle length percentages. Three different regions of interest were detected at 30%, 50% ($CSA_{50}$) and 70% of femur length (measured as the distance between the great trochanter and the mid-patellar point), where 0% represents the mid-patellar point (distal part) and 100% the greater trochanter (proximal part). The transducer was moved slowly in the transverse plane from the medial border of the vastus medialis to the lateral borders of the vastus lateralis, keeping the pressure on the skin as constant as possible. An adjustable guide was used in each acquisition in order to keep the same transverse path (Monti et al., 2020). A generous amount of transmission gel was applied to improve the acoustic contact. Two scans were obtained for each site and the image with the best quality was analysed. CSA measures were obtained by tracing the contours of the quadriceps and vastus lateralis using ImageJ software (1.52v; National Institutes of Health, Bethesda, MD, USA). Quadriceps and vastus lateralis $CSA_{mean}$ were computed by averaging the values at 30%, 50% and 70% femur length of each subject.

**Quadriceps force, rapid force production and activation capacity.** Procedures regarding the *in vivo* muscle function assessment have been described in detail previously (Monti et al., 2021). Briefly, quadriceps force was evaluated during an isometric contraction at 90 degrees of knee flexion using a custom-made knee dynamometer equipped with a load cell. Participants were instructed to push as strongly and as fast as they could for ∼4 s. Visual feedback and loud vocal encouragement were provided. Three trials were recorded, separated by a 1-min rest. The force signal was sampled at 1000 Hz using LabChart software (v.8.13, ADInstrument, Dunedin, New Zealand). The maximum force value reached during these trials was considered the MVC. This value was then divided for quadriceps $CSA_{50}$ to obtain the specific force (force/CSA; $N/cm^2$). The capacity for rapid force production was

evaluated using its time constant (the time required to reach 63% of MVC; $TTP_{63\%}$). Activation capacity, defined as the ability to voluntarily recruit MUs, was evaluated using the interpolated twitch technique. During the MVC procedures, two electrical stimulations (Digitimer DS7AH, Digitimer Ltd, Welwyn Garden, UK) were applied in the proximal and distal region of quadriceps during the MVC plateau, and one approximately 1 s after the end of the contraction. Activation capacity was calculated as previously described (Monti et al., 2021). All the analyses were performed with a custom MATLAB script (version R2021b; The MathWorks, Natick, MA, USA).

### Intramuscular electromyography

**Identification of the motor point.** The vastus lateralis motor point was detected as the location at which the largest muscle twitch was evoked in response to low-intensity percutaneous electrical stimulation using a pen electrode (Botter et al., 2011). Since vastus lateralis usually presents three different motor points (Botter et al., 2011), the central one, located around the mid-thigh, was targeted. Stimulations were induced using a Digitimer DS7AH with an electrical current set at 16 mA (400 V; pulse width: 50 $\mu$s). Once the motor point was identified, the current was reduced to 8–10 mA to verify that it was the most sensitive (i.e. largest twitch) point for stimulation.

**Intramuscular electromyography procedures.** The iEMG signals were recorded using a concentric needle electrode with a diameter of 0.46 mm and a recording area of 0.07 $mm^2$ (S53153; Teca Elite, Natus Medical Inc., Middleton, WI, USA). The iEMG signal was sampled at 40 kHz using the LabChart software (v.8.13, ADInstruments). The iEMG procedures were based on previous works (Jones et al., 2021; Piasecki, Ireland, Stashuk et al., 2016). The needle was inserted diagonally (∼60 degrees) in the vastus lateralis muscle at the motor point. At each new location (see below), participants were asked to perform a very low-intensity contraction (∼5% MVC) and the needle position was slightly adjusted to ensure that the iEMG signal had adequate sharpness (Jones et al., 2021; Piasecki, Ireland, Stashuk et al., 2016). Using an ultrasound device (Mylab70, Esaote, Genoa, Italy), we measured muscle and subcutaneous fat thickness in this location in order to avoid inserting the needle beyond the vastus lateralis deep aponeurosis (Jones et al., 2021). Afterward, participants were asked to perform submaximal isometric contractions at 10% and 25% MVC. Visual feedback was provided. Each contraction lasted 20 s, with 30 s of rest between contractions. Care was taken to maintain a constant needle position during recording. Needle position was slightly

changed after each pair of 10% and 25% contractions by twisting the needle 180 degrees or extracting it by ∼2–3 mm. Recordings were performed at three different depths and two different rotations. A total of 12 contractions (six at 10% and six at 25% MVC) were collected.

**Intramuscular electromyography decomposition and MUP analysis.** DQEMG software was employed to automatically extract MUP trains, representing the electrophysiological activity of individual sampled MUs, from the recorded iEMG signals and perform all subsequent quantitative analyses (Stashuk, 1999a). MUP trains that contained fewer than 35 MUPs or MUPs with signal-to-noise ratios <15 and/or non-physiological shapes were excluded. All extracted MUP trains were reviewed by a trained operator (F.S.) and markers, relative to MUP onset, end, positive peak and negative peaks were adjusted, where appropriate. MUP duration was expressed as the time between the onset and end markers. MUP size was evaluated as MUP area within the MUP duration. MUP complexity was assessed as the number of turns (i.e. a change in MUP direction of at least 20 $\mu$V) (Piasecki, Ireland, Stashuk et al., 2016). The mean inter-discharge interval ($IDI_{mean}$) of the MU firing pattern was also estimated.

**Near fibre electromyography and NMJ transmission stability evaluation.** NF electromyography is based on the electrophysiological assessment of individual muscle fibre potentials (MFPs) generated by fibres located near the electrode recording surface (Piasecki et al., 2021; Sanders et al., 2019). Near-fibre MUPs (NF MUPs) are generated by band-pass filtering MUPs using a second-order low-pass differentiator (Stashuk, 1999b). Compared to near fibres, contributions from (i.e. MFPs generated by) distant fibres will be of lower amplitude and frequency. Therefore, by band-pass filtering MUPs, it is possible to focus on the activity of NFs (i.e. those in close proximity, within ∼350 $\mu$m, to the needle electrode) (Piasecki et al., 2021). All NF MUP trains were visually inspected and NF MUPs containing contaminating MFP components (i.e. activity generated by other MUs) were manually excluded. Only NF MUPs trains with signal-to-noise ratios >15, >34 NF MUPs, and having a NF count (see below) of at least 1 were included. The following parameters were evaluated (Piasecki et al., 2021): (i) NF MUP duration and (ii) NF Count, the number of fibre contributions detected within the NF MUP duration. NMJ transmission stability was evaluated using (iii) NF MUP jiggle and (iv) NF MUP segment jitter, which represent the shape variability (i.e. mean absolute consecutive amplitude differences) and temporal variability (i.e. mean absolute consecutive temporal differences) in consecutive

NF MUPs, respectively (Piasecki et al., 2021; Stålberg & Sonoo, 1994).

## Muscle biopsy

In each participant, a vastus lateralis muscle biopsy (∼150 mg) was collected using a Weil–Blakesley conchotome (Gebrüder Zepf Medizintechnik GmbH & Co. KG, Dürbheim, Germany). Since our intention was to relate the changes in iEMG measures to biopsy outcomes, each biopsy was performed at ∼2 cm from the central motor point, similarly to Aubertin-Leheudre et al. (2019). A distance of ∼2 or 3 cm between the three biopsy sites was maintained to avoid effects of pre-sampling (for further details see, Supporting information, Peer review history). Two millilitres of lidocaine (2%) was injected in the area of the sampling and a small incision of muscle and fascia was performed. Muscle samples were divided into two different parts. The first part, for mRNA analysis, was frozen in liquid nitrogen and stored at −80°C. The second part, for immunohistochemical analysis, was included in optimal cutting temperature (OCT) compound, frozen in isopentane and stored at −80°C. Cryosections were cut with a manual cryostat (Leica CM1850; Leica Microsystems, Wetzlar, Germany), producing 10 $\mu$m-thick sections.

**Immunohistochemistry.** Detection and quantification of denervated and regenerating myofibres were performed by neural cell adhesion molecule (NCAM) and neonatal myosin immunolabelling, respectively, as described: serial cryosections were fixed in pre-chilled methanol at −20°C (NCAM staining only), washed in phosphate-buffered saline (PBS) and then blocked in 10% fetal bovine serum/PBS (both for NCAM and neonatal myosin heavy chain staining). The same cryosections were then labelled (1 h at room temperature) using rabbit polyclonal antibody directed either against NCAM (cat. no. AB5032, Chemicon, Millipore, Milan, Italy) or against laminin (cat. no. L9393, Sigma-Aldrich, St Louis, MO, USA) 1:200 and 1:100 diluted, respectively, in 2% goat serum in PBS. Sections were then rinsed in PBS (3 × 5 min), blocked in 10% goat serum in PBS (10 min at room temperature) and then incubated with goat anti-rabbit IgG Alexa Fluor 594 red fluorescent dye (A-11 012, Thermo Fisher Scientific, Waltham, MA, USA) 1:500 diluted in PBS (1 h at room temperature). After washes, sections stained for NCAM were coverslipped using ProLong Diamond Antifade Mountant with 4′,6-diamidino-2-phenylindole (DAPI) dye (Thermo Fisher Scientific, D1306). The sections devoted to neonatal heavy chain staining were washed in PBS (2 × 5 min), incubated using mouse monoclonal antibody directed against developmental (embryonic and neonatal)-myosin heavy chain (NCL-MHCd,

RRID:AB_563 901, Novocastra, USA) and 1:10 diluted in PBS. Finally, after rinsing $3 \times 5$ min in PBS and blocking in 10% goat serum (20 min), sections were incubated with goat anti-mouse IgG Alexa Fluor 488 green fluorescent dye (Thermo Fisher Scientific, A-11 001) 1:500 diluted in PBS (1 h at room temperature). After washes, sections were coverslipped using ProLong Diamond Antifade Mountant with DAPI dye and observed under a Zeiss microscope (objective $\times 20$; Zeiss Microscopy, Jena, Germany) connected to a Leica DC 300F camera. Serial cryosections belonging to a regenerating rat muscle were used as positive controls, while negative controls were performed by omitting the primary antibodies from sample incubations. NCAM and neonatal myosin-positive fibres were counted on captured images, using ImageJ software (1.52v) and expressed as the number of positive myofibres per total number of myofibres detected in the biopsy area by laminin staining (approximately 400 muscle fibres). NCAM evaluation was performed only at LS0 and LS10 since this was part of a parallel investigation focused only on the unloading period. Analyses were performed blinded to time point.

**ATPase staining.** To evaluate muscle fibre type distribution, conventional techniques were used to stain serial cross-sections for myofibrillar ATPases, as previously described (Carraro et al., 1985). After pre-incubation at pH 4.35, slow-twitch muscle fibres (those possessing a higher ATPase activity) were visualized as dark, while fast-type fibre (or those possessing a low ATPase activity) were lightly stained. To overcome the distortion of obliquely cut or kinked muscle fibres, commonly observed in human muscle biopsies, muscle fibre minimum Feret diameter was manually measured over all the visible fibres (about 200–400 in each section) using ImageJ software (1.52v). Mean, standard deviation and distribution of the diameters were assessed. The variability of muscle fibres diameters was expressed, per each participant and time point, as the coefficient of variation (CV; standard deviation/mean of all the muscle fibres visible on the biopsy section). Fibre type percentage was measured by manually counting all the dark and light observable fibres over the whole biopsy in each section. Fibre type grouping for the quantification of large and very large fibre type clusters ($>10$ or $>20$ fibres, respectively) was evaluated using the method of 'enclosed fibre' (fibre entirely surrounded by fibres of the same histochemical type) (Jennekens et al., 1971). Longitudinally oriented muscle fibres were not considered for analyses. Analyses were performed blinded for time point.

**RNA extraction, sequencing and transcriptomic analysis.** RNA-Seq was employed to study changes in the expression of genes known to be involved in NMJ and skeletal muscle ion channel regulation. The RNA was extracted from ∼10–15 mg of the muscle tissues by Reprocell (Beltsville, MD, USA) using a Norgen Animal Tissue RNA Purification Kit no. 257 (Norgen Biotek Corp, Ontario, CA). The RNA integrity score (RIN) average was 9.21 (range: 7.90–9.86). The concentration of extracted RNA ranged from 34 to 166 ng/$\mu$l per sample. RNA-Seq was performed at the Frederick National Laboratory for Cancer Research Sequencing Facility (NIH). The libraries were made using the TruSeq Stranded Total RNA Library Prep protocol from Illumina (San Diego, CA, USA). This protocol involves the removal of ribosomal RNA (rRNA) using biotinylated, target-specific oligos combined with Ribo-Zero rRNA removal beads. The RNA was fragmented pieces and the cleaved RNA fragments were copied into first-strand cDNA using reverse transcriptase and random primers, followed by second-strand cDNA synthesis using DNA polymerase I and RNase H. The resulting double-strand cDNA was used as the input to a standard Illumina library prep with end-repair, adapter ligation and PCR amplification being performed to provide a library ready for sequencing. Samples were sequenced on a NovaSeq 6000 (Illumina) on an S4 flowcell using paired-end sequencing with read length of 100 bps. The sequencing quality of the reads was assessed using FastQC (v. 0.11.5; https://www.bioinformatics.babraham.ac.uk/projects/fastqc/), Preseq (v. 2.0.3) (Daley & Smith, 2013), Picard tools (v. 2.17.11; https://broadinstitute.github.io/picard/) and RSeQC (v. 2.6.4) (Wang et al., 2012). The samples had 295–419 million pass filter reads (average: 367 million reads per sample) with more than 90% of bases above the quality score of Q30. In addition, Kraken (v. 1.1) (Wood & Salzberg, 2014) was used as a quality-control step to assess microbial taxonomic composition. Reads were trimmed using Cutadapt (v. 1.18) (Martin, 2011) to remove sequencing adapters prior to mapping to the human reference genome hg38 using STAR (v. 2.7.0f) (Dobin et al., 2013) in two-pass mode. Expression levels were quantified using RSEM (v. 1.3.0) (Li & Dewey, 2011) with GENCODE annotation (v. 21) (Harrow et al., 2012). In order to filter out low expression genes, filterByExpr() from package edgeR (v. 3.32.1) (Robinson et al., 2010) was used, resulting in 20,555 genes that passed the filter. Subsequently, quantile normalization was performed using the voom algorithm (Law et al., 2014) from the Limma R package (v. 3.46.0) (Smyth, 2004), followed by empirical Bayesian smoothing of standard errors to assess case *vs.* control differentially expressed genes adjusted for subject ID (paired analysis). Based on *P*-values derived from this analysis, all measured genes were ordered by $-\log 10$(*P*-value); this ordered list of genes was analysed to assess enrichment of a pre-selected set of pathways associated with NMJs and ion channels in skeletal muscle. Gene set enrichment analysis (GSEA) was performed

using package fgsea (v. 1.20.0) (Korotkevich et al., 2021). A customized implementation in R was developed in-house to add robustness to the GSEA analysis; since GSEA's enrichment estimates (and statistical significance) are stochastic, our software embeds GSEA in a Monte Carlo algorithm that performs 1000 iterations and chooses significant pathways based on the total statistical ensemble. Leading edge genes shown in Fig. 5 were those selected by GSEA in more than 80% of the iterations.

### Blood sampling and circulating biomarkers evaluation

Blood samples were obtained from the medial cubital vein in Gel Clot Activator tubes (368,969, BD Diagnostic, Oxford, UK) before the *in vivo* muscle function evaluation and were then centrifuged (CN45, Eurotek, Orma, Milan, Italy) at 1100 *g* for 10 min to separate serum from the other blood components. Samples were aliquoted and consequently stored at −80°C until analysis.

In order to investigate NMJ damage, serum CAF concentration was evaluated. Serum CAF concentration was measured using a commercially available enzyme-linked immunosorbent assay (ELISA) kit (Human Agrin SimpleStep ELISA, Ab216945, Abcam, Cambridge, UK) following the manufacturer's instructions. A microplate ELISA reader (Infinite F50, Tecan Trading AG, Männedorf, Switzerland) was employed to read the absorbance at 450 nm. CAF concentrations were obtained by interpolation with a standard curve and corrected for sample dilution (Marcolin et al., 2021; Monti et al., 2021).

Single molecule array (SIMOA) analysis was also performed to evaluate neurofilament light chain concentration, a well-established biomarker of axonal damage, previously employed both in aging and neurological disorders scenarios (Khalil et al., 2018; Pratt et al., 2022). The samples were submitted to the SIMOA service offered by Wieslab AB (a Svar Life Sci company, Lundavägen, Malmo, Sweden) in accordance with Good Laboratory Practice (GLP) principles. SIMOA analysis was performed on a Simoa HD-X Analyzer (PN 10041537) supplied by Quanterix (USA) in agreement with standard protocol suggested by Quanterix. In particular, serum samples were diluted 1:4 and measurements obtained in double replicates. The kit used for the reported analysis was Simoa NF-light Advantage Kit HD-1/HD-X (item 103186; lot 502845).

### Statistical analysis

*A priori* power analysis was performed based on changes in CAF concentration to determine the required sample size. For an effect size calculated based on our previous work (Monti et al., 2021), a required power $(1 - \beta)$ of 0.9 and an error $\alpha = 0.05$, the total required sample size was 11 subjects. Therefore, the 12 participants recruited in this work represented an appropriate sample, considering potential study drop-outs. Normality of *in vivo* muscle size and function data, circulating biomarker concentrations, fibre diameter variability, fibre type and markers of denervation were assessed through the Shapiro–Wilk normality test and visual inspection of Q–Q plots. All the considered parameters passed normality tests. For all these variables, one-way repeated-measures ANOVA (regular or mixed-effects, depending on missing values) with Tukey's *post hoc* test were performed to determine whether differences among the different time points were present. For all ANOVAs, sphericity was tested with Mauchly's test and when the assumption of sphericity was violated, the Greenhouse–Geisser correction was applied. NCAM and slow fibre type grouping datasets did not pass normality tests and were evaluated with a non-parametric Wilcoxon's test or Friedman's test, respectively. Gene expression analysis was performed separately on two sets of genes identified as relevant for NMJs, based on 12 pathways extracted from the Molecular Signature Database MSigDB (v. 7.4) (Subramanian et al., 2005), and skeletal muscle ion channels (Jurkat-Rott & Lehmann-Horn, 2004) (see Supporting information Data S1, Data S2 and Data S3). For each gene set, we performed principal components analysis (PCA) and differential GSEA (see details above). For the analysis of iEMG data, we performed generalized linear mixed effect models (fixed effect: time; cluster variable: subject), as multiple MUs were recorded from each participant (Yu et al., 2022). The family of the distribution employed was the gamma or inverse Gaussian distribution depending on each variable, with different associated link functions (Table 1). The function used to compare the different models was the Bayesian information criterion (BIC). For models with similar minimal BIC, the canonical link function of the distribution (inverse function $(1/y)$ and inverse squared $(1/y^2)$ for gamma distribution and inverse Gaussian, respectively) was chosen. *Post hoc* comparisons were performed with the Holm correction. Statistical significance was set at $P < 0.05$. Jamovi software (version 2.2, Sydney, Australia) was used to perform statistical analysis, while GraphPad Prism (version 8.00; GraphPad Software, San Diego, CA, USA) was employed to create all of the graphs.

### Results

No side effects regarding biopsy procedures, ULLS exposure or the AR phase were reported. One participant dropped out after baseline measures for personal reasons not related to the study and another one did not perform iEMG measures at 25% MVC at LS10 due to pain

experienced during contractions. Two other participants decided to not undergo muscle biopsy at AR21 for personal reasons.

### *In vivo* muscle morphology and function

First, we evaluated whether the ULLS period resulted in a significant decline in muscle size and function and whether the AR phase was effective in counteracting this loss. For quadriceps ($P < 0.001$; $\eta_p^2 = 0.849$) and vastus lateralis ($P < 0.001$; $\eta_p^2 = 0.839$) $CSA_{mean}$, a main effect of time was observed (Fig. 1). $CSA_{mean}$ was significantly reduced for both quadriceps ($P = 0.007$) and vastus lateralis ($P = 0.038$) at LS10, and subsequently increased and at AR21 they were significantly higher compared to LS10 (quadriceps: $P < 0.001$; vastus lateralis: $P < 0.001$). Interestingly, $CSA_{mean}$ at AR21 was higher than LS0 and LS10 (quadriceps: $P < 0.001$; vastus lateralis: $P < 0.001$). All muscle function parameters were significantly influenced by time (Fig. 1): MVC ($P < 0.001$;

$\eta_p^2 = 0.842$), specific force ($P < 0.001$; $\eta_p^2 = 0.724$); $TTP_{63\%}$ ($P < 0.001$; $\eta_p^2 = 0.645$) and activation capacity ($P = 0.004$; $\eta_p^2 = 0.514$). A significant decline in MVC was observed at LS10 ($P < 0.001$), followed by full recovery at AR21 ($P < 0.001$) but not at AR10, which was still reduced compared to LS0 ($P = 0.004$). The specific force (LS0 *vs.* LS10: $P < 0.001$; LS10 *vs.* AR21: $P = 0.002$) and activation capacity (LS0 *vs.* LS10: $P = 0.02$; LS10 *vs.* AR21: $P < 0.001$) followed the same pattern, with a difference also between LS0 and AR21 for specific force ($P = 0.033$). The $TTP_{63\%}$ was significantly increased at LS10 ($P = 0.004$) and returned to baseline levels at AR21 ($P = 0.003$).

### MUP properties and NMJ transmission stability

To investigate whether changes in muscle morphology and function were accompanied by electrophysiological changes, MUP properties and NMJ transmission stability were assessed with iEMG. MUPs from 814 (24.77 (12.8)

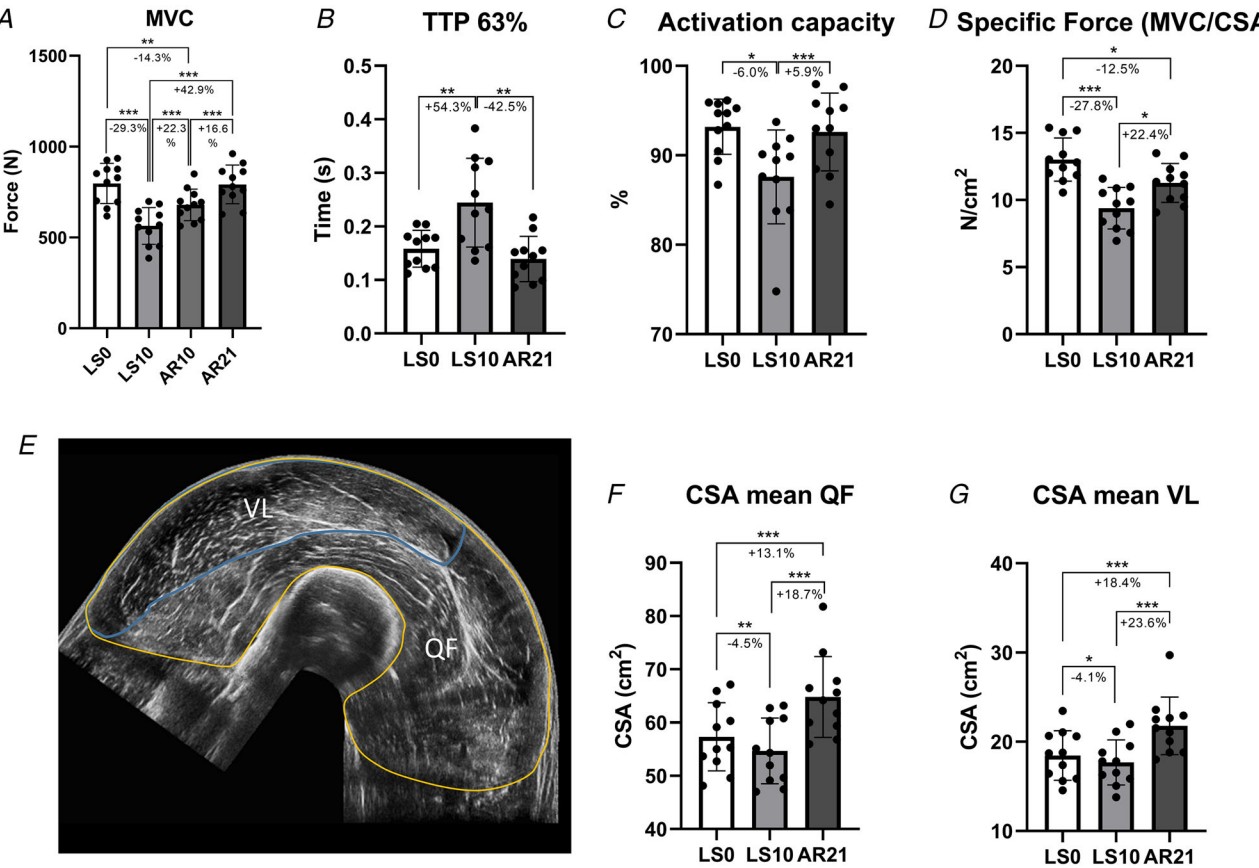

**Figure 1. Changes in the *in vivo* muscle morphology and function following 10 days of unilateral lower limb suspension (LS10) and 10 days (AR10) and 21 days of active recovery (AR21)**
Statistical analysis was performed using regular repeated-measures one-way ANOVAs. Results are shown as mean and standard deviation. Maximum voluntary contraction (*A*); time needed to reach the 63% of the MVC ($TTP_{63\%}$) (*B*); activation capacity (*C*); specific force (*D*); representative quadriceps femoris (QF) and vastus lateralis (VL) panoramic ultrasound image (*E*); mean cross-sectional area (CSA) for QF (*F*) and VL (*G*). *$P < 0.05$; **$P < 0.01$; ***$P < 0.001$. LS0: baseline data collection. [Colour figure can be viewed at wileyonlinelibrary.com]

on average per participant at each time point) and 1212 MUs (21.5 (11.96) on average), sampled at 10% and 25% MVC, respectively, were analysed. NF MUPs from 495 and 709 MUs, sampled at 10% (15.0 (10.26) on average) and 25% (36.73 (16.0) on average) MVC, respectively, were analysed. Results for MUP and NF MUP characteristics are shown in Figs 2 and 3, respectively. No spontaneous activity (i.e. fibrillation or fasciculation potentials) was observed at any of the time points (data not shown). An increased $IDI_{mean}$, reflecting a lower MU firing rate, was observed at LS10 for both 10% ($P < 0.001$) and 25% ($P = 0.03$) MVC, while $IDI_{mean}$ was decreased at 10% at AR21 compared to LS0 ($P = 0.008$) and LS10 ($P < 0.001$). MUP size (MUP area) was not affected by suspension but it was reduced at AR21 with respect to LS0 (10% MVC: $P < 0.001$; 25% MVC: $P < 0.001$) and LS10 (10% MVC: $P = 0.008$). MUP duration followed the same pattern (LS0 *vs.* AR21, 10% MVC: $P < 0.001$, 25% MVC: $P < 0.001$; LS10 *vs.* AR21, 10% MVC: $P < 0.001$, 25% MVC: $P < 0.001$). At 25% MVC, MUP complexity (evaluated as the number of turns) was found to be increased at LS10 ($P < 0.001$) and restored at AR21 ($P < 0.001$). NF count was lower at LS0 compared to LS10 and AR21 both at 10% (LS0 *vs.* LS10: $P = 0.007$; LS0 *vs.* AR21: $P = 0.014$) and 25% MVC (LS0 *vs.* LS10: $P = 0.001$; LS0 *vs.* AR21: $P = 0.001$). NF MUP duration was greater at LS10 compared to LS0 at 25% MVC (25%

MVC: $P = 0.032$), with a trend also at 10% ($P = 0.06$); and had a further increase at AR21 (LS0 *vs.* AR21, 10% MVC: $P < 0.001$, 25% MVC: $P < 0.001$; LS10 *vs.* AR21, 10% MVC: $P < 0.001$, 25% MVC: $P = 0.032$). Regarding NMJ transmission stability, NF MUP jiggle was unmodified at LS10 compared to LS0 but decreased at 10% MVC at AR21 compared to the other two time points (LS0 *vs.* AR21: $P = 0.002$; LS10 *vs.* AR21: $P = 0.001$). Finally, NF MUP segment jitter was not influenced by the interventions. Additional information on the iEMG data analysis is shown in Table 1.

## NMJ and ion channels transcriptomic profile

In view of the observed changes in iEMG parameters, we investigated molecular events potentially connected with changes in synaptic transmission and electrical activity. In particular, we performed RNA-Seq and found a significant shift in the overall NMJ transcriptomic profile (across 290 NMJ-associated genes) comparing LS0 and LS10, with these changes restored at AR21, as evidenced by PCA (Fig. 4*A*). Interestingly, PCA showed greater variance at LS10, indicating a heterogeneous adaptive response between participants to ULLS. The volcano plots illustrate those 290 NMJ genes compared pairwise between time points (Fig. 4*B–D*). In each of these volcano plots, the *y*-axis shows $-\log_{10}(P\text{-value})$; reference lines

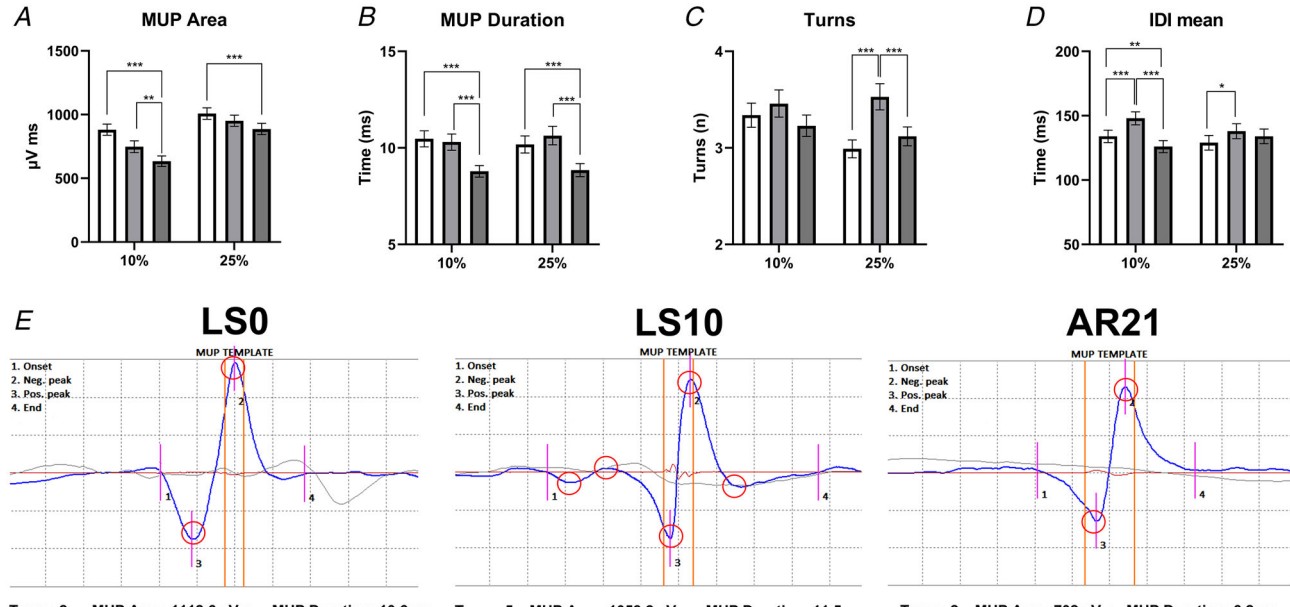

**Figure 2. Changes in MUP properties obtained using intramuscular electromyography after 10 days of unilateral lower limb suspension (LS10) and 21 days of active recovery (AR21)**
Statistical analysis was performed using generalized linear mixed effect models. Results are shown as estimated marginal mean and standard error. Motor unit potential (MUP) area (*A*); MUP duration (*B*); MUP turns (*C*); mean inter-discharge interval ($IDI_{mean}$) (*D*); and representative MUP templates at each time point (*E*). MUP area represents the area under the MUP waveform displayed in blue; MUP Duration was computed as the time between markers 1 and 4; each turn is highlighted by a red circle. *$P < 0.05$; **$P < 0.01$; ***$P < 0.001$. LS0: baseline data collection.

**Table 1. Details of the generalized linear mixed effect models performed for each variable**

| Parameter | Distribution | Link function | Estimate | 95% confidence intervals | P |
|---|---|---|---|---|---|
| MUP area 10% ($\mu$V ms) | Inverse gaussian | Identity | 734.4 | 662 to 807.3 | <0.001 |
| MUP area 25% ($\mu$V ms) | Inverse gaussian | Identity | 949 | 869 to 1028.8 | <0.001 |
| MUP duration 10% (ms) | Gamma | Inverse | 0.10213 | 0.09545 to 0.10188 | <0.001 |
| MUP duration 25% (ms) | Gamma | Inverse | 0.10171 | 0.094 to 0.10942 | <0.001 |
| MUP turns 10% (number of turns) | Inverse gaussian | Inverse squared | 0.08971 | 0.07874 to 0.10068 | 0.142 |
| MUP turns 25% (number of turns) | Inverse gaussian | Inverse squared | 0.09821 | 0.0874 to 0.10907 | <0.001 |
| IDI$_{mean}$ 10% (ms) | Inverse gaussian | Identity | 135.86 | 127.05 to 144.67 | <0.001 |
| IDI$_{mean}$ 25% (ms) | Inverse gaussian | Identity | 133.78 | 123.13 to 144.4 | 0.033 |
| NF MUP duration 10% (ms) | Gamma | Inverse | 0.4405 | 0.3871 to 0.49394 | <0.001 |
| NF MUP duration 25% (ms) | Gamma | Inverse | 0.4482 | 0.4004 to 0.49497 | <0.001 |
| NF count 10% (number) | Inverse gaussian | Inverse | 0.8546 | 0.792 to 0.9142 | 0.005 |
| NF count 25% (number) | Inverse gaussian | Inverse | 0.8198 | 0.763 to 0.8767 | <0.001 |
| NF MUP jiggle 10% (%) | Inverse gaussian | Inverse squared | 9.132 | 8.175 to 10.088 | <0.001 |
| NF MUP jiggle 25% (%) | Inverse gaussian | Inverse squared | 13.5902 | 11.661 to 15.52 | 0.948 |
| NF MUP jitter 10% ($\mu$s) | Gamma | Inverse | 0.03635 | 0.03254 to 0.04016 | 0.152 |
| NF MUP jitter 25% ($\mu$s) | Gamma | Inverse | 0.028 | 0.02504 to 0.03102 | 0.409 |

The overall estimate and *P*-value of the model are reported in this table, while *P*-values of the time-point comparison are presented in the text. IDImean: mean inter-discharge interval; MUP: motor unit potential; NF MUP: near fibre motor unit potential.

display the false-discovery rate (FDR) *q*-values 0.05 and 0.01. All the volcano plots for NMJ genes were fairly symmetrical, suggesting both upregulation and down-regulation of NMJ genes in response to the interventions. With unloading, 95 NMJ genes were differentially

regulated at *q* < 0.05 (70 genes at *q* < 0.01) (Fig. 4*B*), including some encoding ACh receptor subunits, neuro-trophins (such as *NT4* and *GDNF*), Homer proteins and other key regulators of the NMJ (neuregulins, the epidermal growth factor receptor (ErbB) and Wnts

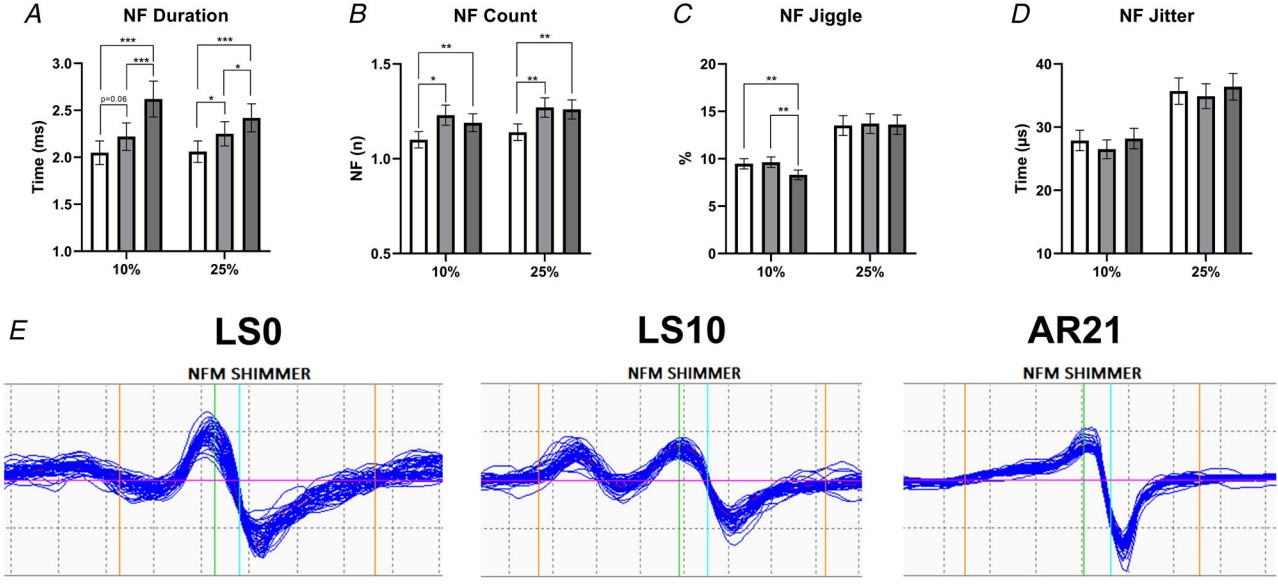

**Figure 3. Changes in near fibre (NF) electromyography outcomes following 10 days of unilateral lower limb suspension (LS10) and 21 days of active recovery (AR21)**
Statistical analysis was performed using generalized linear mixed effect models. Results are shown as estimated marginal mean and standard error. Near fibre motor unit potential (NFM) duration (*A*); NF count (*B*); NFM jiggle (*C*); NFM segment jitter (*D*); and representative NFM shimmers at each time point (*E*). *P < 0.05; **P < 0.01; ***P < 0.001. LS0: baseline data collection.

family). Most of these differentially expressed genes (a total of 114 genes at $q < 0.05$ and 81 genes at $q < 0.01$) showed an opposite trend at AR21 (Fig. 4C). Comparing LS0 and AR21, differences were much less pronounced: only 24 genes were significantly differentially expressed at $q < 0.05$ (7 genes at $q < 0.01$) with also more limited log fold changes (Fig. 4D). NMJ RNA-Seq dataset is available (see Supporting information, Data S1).

GSEA showed changes in some NMJ-related gene sets at LS10 and AR21, but not comparing LS0 and AR21 values (see Supporting information, Data S3). Differences in NMJ leading-edge genes based on GSEA are presented in Fig. 5. In the top quadrant of the figure, a group of genes that were downregulated following unloading and upregulated after the AR period is shown, with *COLQ* (a collagen-tail subunit of acetylcholinesterase) being the most responsive gene. In contrast, in the bottom right quadrant, genes that increased their expression with ULLS and decreased it at AR21 are presented, including two ACh receptor subunits (*CHRNA1* and *CHRND*).

In addition, we analysed 33 genes associated with skeletal muscle ion channels due to their key role in the maintenance of membrane resting conductance and potential and in the propagation of electrical impulses along the sarcolemma (Jurkat-Rott & Lehmann-Horn, 2004). Analogously to NMJ genes, PCA analysis showed similar overall gene expression at LS0 and AR21, while at LS10 it showed a separated and heterogeneous cluster (Fig. 6A). As illustrated in the volcano plot, differences at LS10 appeared asymmetrical and biased towards LS0 (Fig. 6B), suggesting an overall down-regulation of the ion channel gene set, with voltage-gated potassium channel genes that seem particularly affected by unloading. Differently, the volcano plot comparing LS10 and AR21 was symmetrical: expression of different potassium channel genes was restored at AR21 but other genes encoding, for instance, some chloride and calcium channels subunits had reduced expression (Fig. 6C). Only very small differences were observed in the comparison between LS0 and AR21 results, with few genes remaining downregulated (Fig. 6D). The ion channel RNA-Seq dataset is available (Supporting information Data S2). Overall, these findings suggest that genes of NMJs and skeletal muscle ion channels underwent adaptive changes between LS0 and LS10, which are mostly reversed at AR21.

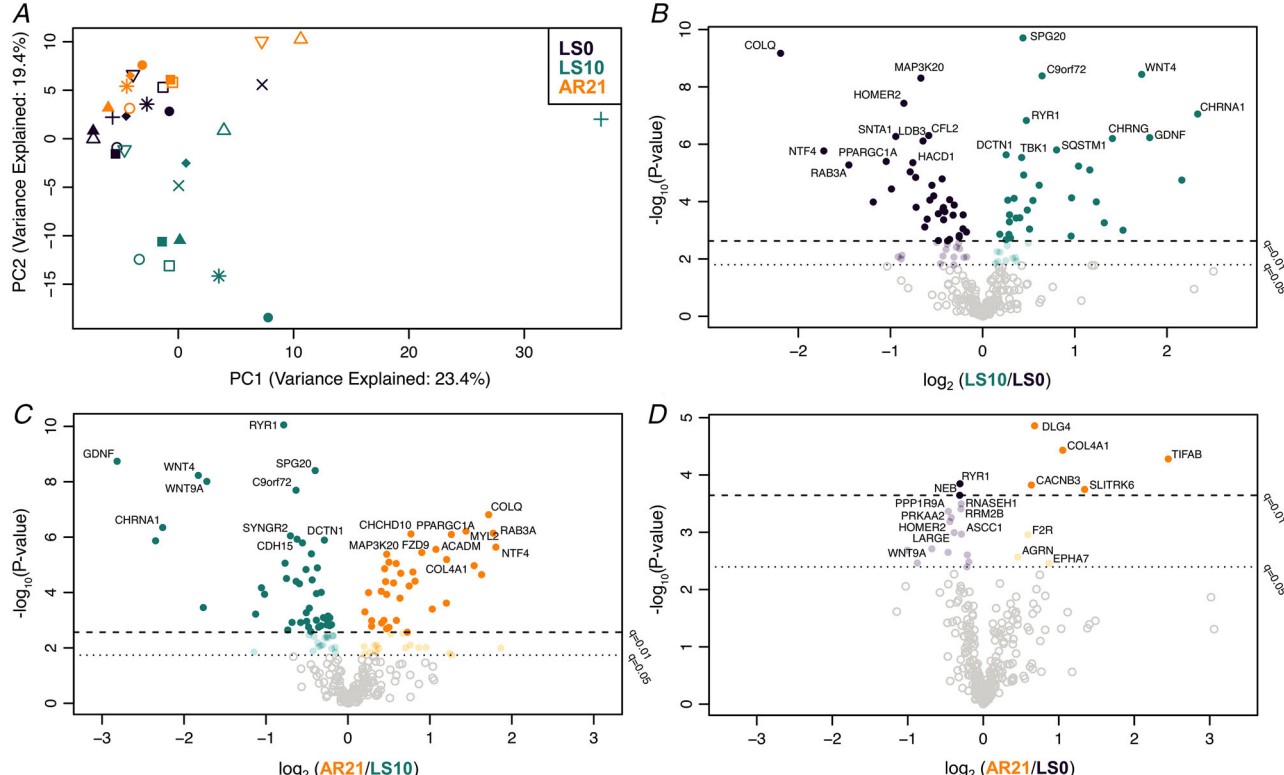

**Figure 4. Changes in the expression of 290 genes related to NMJ during unloading and active recovery as determined by RNA-Seq**
(A) Principal component analysis plot showing a shift in the overall expression of genes involved in neuromuscular junction (NMJ) regulation with 10 days of unilateral lower limb suspension (LS10) and 21 days of active recovery (AR21). *B–D*, volcano plots displaying differentially expressed NMJ coding genes in baseline (LS0) *vs*. LS10 (B), LS10 *vs*. AR21 (C) and LS0 *vs*. AR21 (D). Samples are coloured based on time points.

## Muscle fibre diameter variability, type and markers of regeneration/denervation

Further investigating potential mechanisms underpinning the alterations in MUP properties observed with ULLS, we evaluated several parameters derived from the immuno-

histochemical and ATPase staining analysis (Fig. 7) and some circulating biomarkers (Fig. 8). No differences were detected for fibre diameter variability for either slow or fast fibres during the interventions. Fibre type was not affected by ULLS, but slow fibre percentage increased (and consequently, fast decreased) with AR (LS0 vs. AR21:

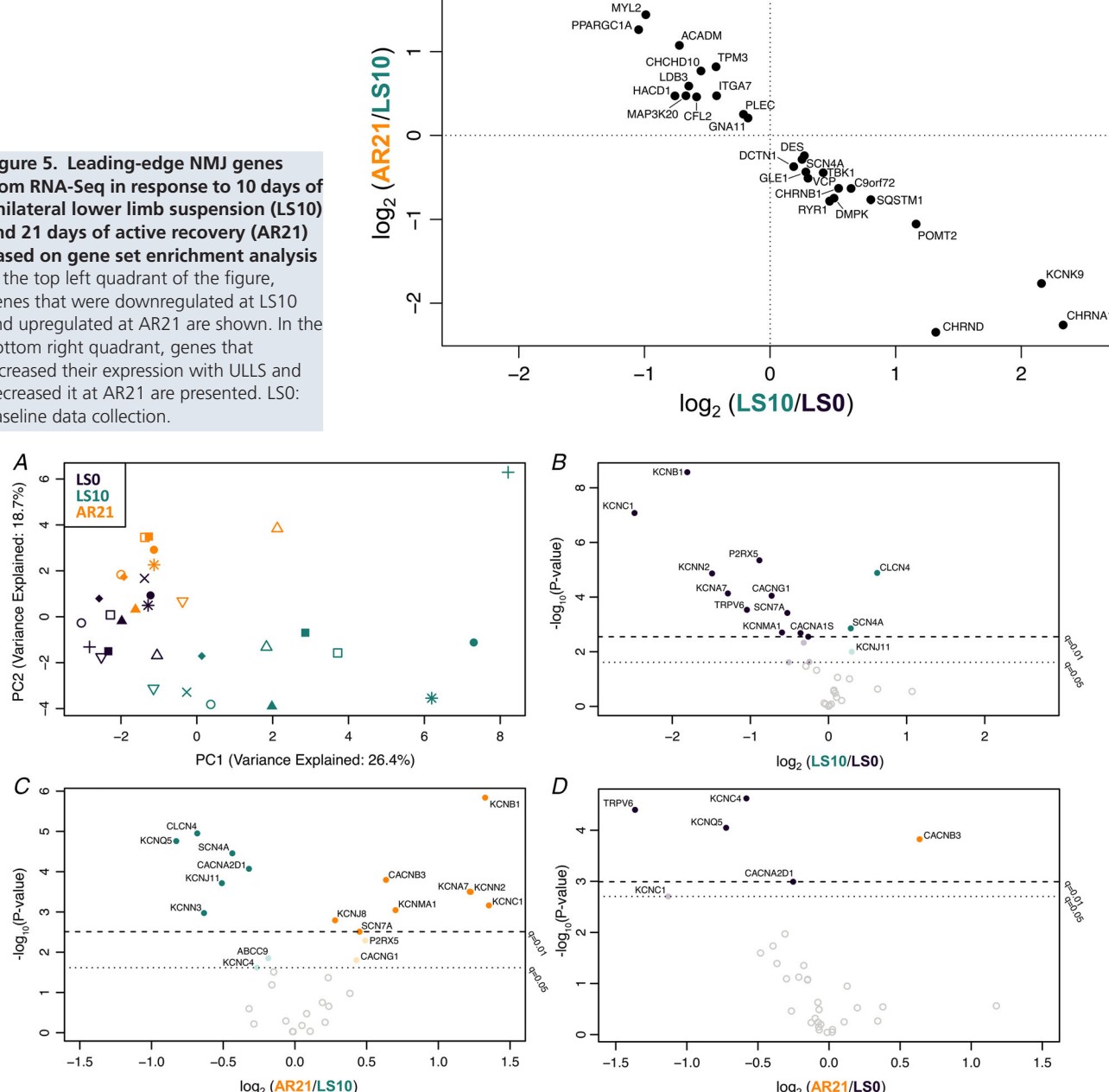

**Figure 5. Leading-edge NMJ genes from RNA-Seq in response to 10 days of unilateral lower limb suspension (LS10) and 21 days of active recovery (AR21) based on gene set enrichment analysis**
In the top left quadrant of the figure, genes that were downregulated at LS10 and upregulated at AR21 are shown. In the bottom right quadrant, genes that increased their expression with ULLS and decreased it at AR21 are presented. LS0: baseline data collection.

**Figure 6. Changes in expression of 33 genes related to skeletal muscle ion channels during unloading and active recovery as determined by RNA-Seq**
(A) Principal component analysis plot showing a shift in the overall expression of genes regulating ion channels with 10 days of unilateral lower limb suspension (LS10) and 21 days of active recovery (AR21). B–D, volcano plots displaying differentially expressed ion channel-encoding genes in baseline (LS0) vs. LS10 (B), LS10 vs. AR21 (C) and LS0 vs. AR21 (D). Samples are coloured based on time points.

$P = 0.004$; LS10 *vs.* AR21: $P < 0.001$). Regarding changes of denervation markers with unloading, no flat-shaped/angulated fibres were observed at any time point (data not shown) and no changes were shown in the percentage of fibre type grouping both for slow and fast fibres. However, as a reflection of the shift towards a slow phenotype in fibre type at AR21, the percentage of fibre type grouping for fast fibre decreased (LS0 *vs.* AR21: $P = 0.034$; LS10 *vs.* AR21: $P = 0.021$). In addition, no regenerating neonatal myosin-positive fibres were found (data not shown), while the percentage of NCAM-positive fibres increased at LS10 ($P = 0.031$; Fig. 8). Evidence of initial axonal damage with unloading was shown by increased neurofilament light chain level at LS10 ($P = 0.001$), subsequently restored at AR21 ($P = 0.001$). Finally, we evaluated serum CAF concentration, a well-established biomarker of NMJ molecular stability. CAF is released in the circulation upon neurotrypsin-induced cleavage of agrin, a proteoglycan that stabilizes the synaptic structures (Stephan et al., 2008). Therefore, a higher concentration

of this circulating biomarker reflects an increased NMJ molecular instability (Drey et al., 2013; Marcolin et al., 2021; Monti et al., 2021). Importantly, our results showed increased CAF concentration increased at LS10 compared to LS0 ($P = 0.038$), with no differences between the other time points.

## Discussion

The aim of this study was to investigate the neuromuscular changes induced by a short period of lower limb unloading (10 days of ULLS) followed by active recovery (21 days of resistance exercise), with a particular focus on changes in NMJ transmission stability, motor unit potential (MUP) characteristics and underlying molecular mechanisms. The main findings of this study are: (i) human NMJs are functionally resilient despite molecular destabilization induced by a short period of unloading, at least at relatively low contraction intensity (activity of low-threshold, slow-type MUs); (ii) unloading

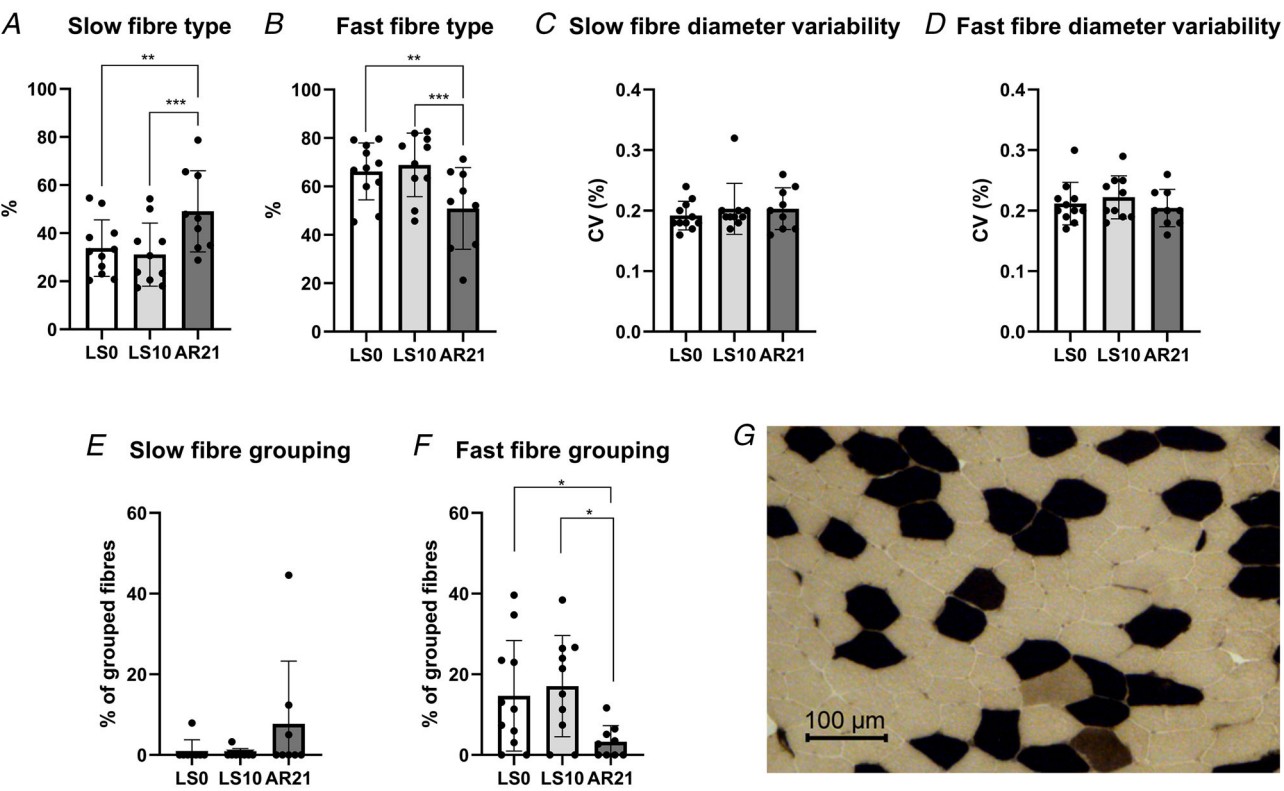

**Figure 7. Changes in parameters obtained by ATPase staining following 10 days of unilateral lower limb suspension (LS10) and 21 days of active recovery (AR21)**
Statistical analysis was performed using mixed-effects repeated-measures one-way ANOVAs (*A–D, F*) and Friedman's test (*E*). Results are shown as mean and standard deviation. Percentage of slow fibre (*A*); percentage of fast fibre (*B*); fibre diameter variability in slow fibre (*C*); fibre diameter variability in fast fibre (*D*); percentage of grouped slow fibre (*E*); percentage of grouped fast fibre (*F*); and a representative ATPase staining (*G*). We performed pH 4.35 ATPase, and hence fibres possessing a high ATPase activity (oxidative) are dark, while fibres possessing a low ATPase activity (glycolytic) are clear. Fibres possessing an intermediate metabolism are brown. *$P < 0.05$; **$P < 0.01$. LS0: baseline data collection.

causes alterations in MUP and near fibre (NF) MUP characteristics; and (iii) the AR period reversed most of these neuromuscular changes.

## Muscle structure and function and NMJ alterations with unloading

The 10-day ULLS intervention resulted in marked impairments of quadriceps muscle isometric force and capacity for rapid force production ($TTP_{63\%}$). These functional impairments were accompanied by a 4.5% loss in quadriceps size, in line with a previous study reporting a ∼5% decrease in quadriceps CSA after 14 days of ULLS (de Boer et al., 2007).

Our results show a modest but significant increase in CAF concentration at the end of the unloading period suggesting that 10 days of unilateral limb unloading was sufficient to induce initial NMJ molecular instability. The increase in CAF concentration in the present study appears less pronounced than that observed after 10-day bed rest (5.4% *vs.* 19.2%, respectively) (Monti et al., 2021), most likely due to the lower muscle mass undergoing unloading (one leg in ULLS *versus* whole body in bed rest). NMJ molecular alterations with unloading were further suggested by our RNA-Seq results, showing changes in the expression of many genes known to be involved in NMJ regulation. Among these, *CHRNA1*, *CHRNB1*, *CHRND* and *CHRNG*, coding for the α1, β, δ (all normally expressed in adult skeletal muscle) and γ (a fetal form) subunits of the ACh receptor, were found to be upregulated between baseline and after ULLS. In normal conditions, ACh receptor expression is limited to the synaptic region and the γ subunit is maintained only at very low expression levels. However, it is well-established that with denervation and motoneuron dysfunction muscle fibres re-express the fetal γ subunit

and in addition, ACh receptors are re-expressed over their whole surface (Hughes et al., 2006; Soendenbroe et al., 2021). Thus, the upregulation of these genes may represent the onset of these processes and a first sign of an altered innervation pattern. In addition, downregulation of the gene encoding acetylcholinesterase (*ACHE*) and its non-catalytic subunit (*COLQ*) suggests an impairment in the process of ACh elimination from the synaptic cleft, potentially leading to excessive sodium influx through ACh receptor channels which could in turn damage the postsynaptic membrane (Hughes et al., 2006; Tintignac et al., 2015). Interestingly, our findings are in agreement with a recent meta-analysis based on previously published transcriptomics datasets, detecting *CHRNA1*, *CHRND* and *COLQ* among the top inactivity-responsive genes (Pillon et al., 2020). In addition, genes regulating Homer proteins, which are known to be involved in the expression of synaptic genes via the calcineurin-nuclear factor of activated T-cells (NFATc1) signalling pathway (Nishimune et al., 2014), were downregulated, as previously reported with both short-term (10 days) (Monti et al., 2021) and long-term (60 days) bed rest (Salanova et al., 2011). Changes in the expression of some genes regulating neuregulins (*NRG2*), neurotrophins (*NT4* and *GDNF*) and the ErbB (*ERBB2* and *ERBB3*) and Wnts (*WNT4* and *WNT9A*) family, which can contribute to NMJ maintenance in different ways (Bloch-Gallego, 2015; Gonzalez-Freire et al., 2014; Nishimune et al., 2014; Tintignac et al., 2015), further confirm initial NMJ molecular instability. Although we did not investigate changes in NMJ morphology, overall our findings highlighted initial and early NMJ instability at the molecular level with unloading, in agreement with previous research conducted in animal models (Deschenes & Wilson, 2003; Fahim, 1989; Fahim & Robbins, 1986; Pachter & Eberstein, 1984). The present

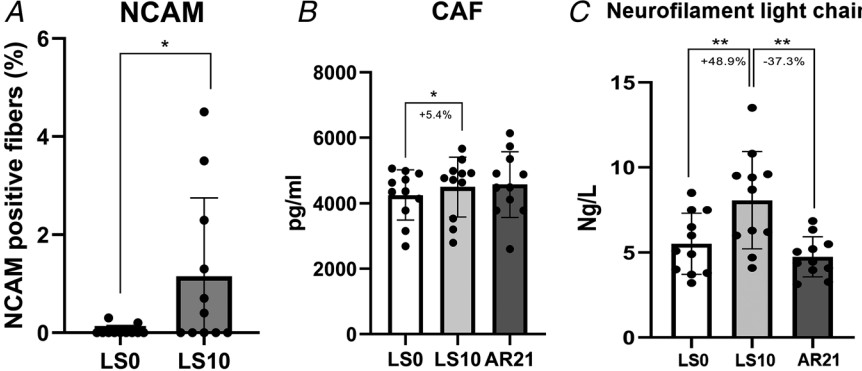

**Figure 8. Changes in neural cell adhesion molecule (NCAM) positive fibre and circulating biomarker following 10 days of unilateral lower limb suspension (LS10) and 21 days of active recovery (AR21)**
Statistical analysis was performed using Wilcoxon's test (*A*) and regular repeated-measures one-way ANOVAs (*B* and *C*). Results are shown as mean and standard deviation. Neural cell adhesion molecule (NCAM) positive fibre (*A*); C-terminal agrin fragment concentration (CAF) (*B*); and neurofilament light chain concentration (*C*). *$P < 0.05$; **$P < 0.01$. LS0: baseline data collection.

study did not aim to investigate the causes of NMJ damage with ULLS, but possible players could be (i) oxidative damage due to mitochondrial dysfunction (Rygiel et al., 2016), (ii) inflammation (Gonzalez-Freire et al., 2014), and (iii) altered calcium transient pattern due to decreased nerve activity (Schiaffino & Serrano, 2002).

However, contrary to our hypothesis, NMJ transmission stability, assessed through the electrophysiological evaluation of NF MUP jiggle and segment jitter, was unchanged after ULLS. This finding supports the body of literature highlighting that the NMJ is a functionally robust structure able to compensate for different insults and stresses (Plomp, 2017; Robbins, 1992; Wood & Slater, 2001). The safeguarding of NMJ transmission is regulated by its safety factor (Wood & Slater, 2001). This term is based on the concept that the number of quanta of ACh released and the induced depolarization of the postsynaptic membrane (endplate potential) per nerve impulse are much greater than that required to generate an action potential in the muscle fibre (Wood & Slater, 2001). This phenomenon makes NMJ transmission an extremely reliable process under various physiological and pathophysiological conditions. Our CAF concentration and transcriptomic profile results indicate an initial NMJ molecular instability after the ULLS intervention that could potentially have functional implications, affecting quantal release and/or the endplate potential. However, these stresses were probably insufficient for reducing the NMJ safety factor, and thus unlikely to impair NMJ transmission stability. Moreover, it is worth noting that the NMJ is known to exhibit remarkable compensatory plasticity that can counterbalance the safety factor reduction in ageing and disease scenarios (Plomp, 2017; Robbins, 1992), with similar positive remodelling that may occur also with unloading. In support of this concept, in animal studies investigating neurotransmitter handling with disuse/unloading, unchanged or increased release of quanta of ACh was previously found (Robbins & Fischbach, 1971; Snider & Harris, 1979; Tsujimoto & Kuno, 1988).

Although we showed no differences in NMJ function after 10-day ULLS, we cannot exclude that different results may be observed with longer periods and/or more extreme forms of disuse/unloading, such as casting or dry-immersion. In fact, the only study that investigated these aspects in humans based on 28-day cast immobilization found increased jitter in the soleus of healthy subjects ($n = 6$; aged 21 to 48 years) (Grana et al., 1996). This observation prompts us to hypothesize that human NMJs are only transiently resilient from a functional perspective to muscle unloading and that NMJ functional impairment is likely to occur beyond 10 days of unloading. Finally, contraction intensity should also be considered. In our study, participants were asked to perform contractions up to 25% MVC and, therefore,

we acknowledge that our findings might be limited to relatively low-intensity contractions. Indeed, we recorded the activity only of low-threshold, slow-type MUs and these may have been differently affected by unloading compared to high-threshold, fast-type MUs.

## Changes in MUP characteristics with unloading and underlying mechanisms

We report that MU electrophysiological properties were altered following ULLS. To the best of our knowledge, only a few pioneering studies investigated the effects of immobilization on human MUs and MUP characteristics, showing reduced MU firing rates in small hand muscles (specifically, the adductor pollicis and the first dorsal interosseous) following disuse during submaximal (Seki et al., 2001) and maximal contractions (Duchateau & Hainaut, 1990; Seki et al., 2007). Similarly, in our study we observed increased $IDI_{mean}$ values, reflecting decreases in mean MU firing rates, potentially explained by reduced ability to activate MUs, as suggested by declines in the activation capacity (evaluated using the interpolated twitch technique) after ULLS. Moreover, our observation of an increase in neurofilament light chain levels after 10 days of ULLS, indicating axonal damage, seems consistent with the observed decrease in MU firing rates. These findings suggest an early impairment upstream to the NMJ (central nervous system and/or motoneuron) that could potentially contribute to the marked loss of muscle function observed.

In addition, our results showed increased MUP turns, NF MUP duration and NF count, overall indicating a more complex MUP and NF MUP shape. Interestingly, employing iEMG, similar changes in MUP and NF MUP properties were previously observed with ageing (Piasecki, Ireland, Stashuk et al., 2016; Piasecki, Ireland, Coulson et al., 2016; Kirk et al., 2019) and neuromuscular diseases such as neuropathies (Allen et al., 2015; Estruch et al., 2019; Gilmore et al., 2020). MUP and NF MUP features are the result of the summation of MFPs generated by the fibres of its MU. Thus, alterations in MUP and NF MUP properties are a reflection of the temporal dispersion of propagating muscle fibre action potentials (MFAPs) at the recording point (i.e. needle axial location) (Piasecki et al., 2021). Additionally, declines in both axonal (due to demyelination of axonal twigs) and muscle fibres conduction velocities could contribute to magnify previously undetectable differences in temporal dispersion of propagating MFAPs at the recording point. A potential mechanism that could explain these differences in conduction times along axonal branches and/or muscle fibres are denervation/reinnervation processes (Piasecki et al., 2021). We showed that unloading may have caused some initial and partial processes of denervation, as suggested for instance by upregulation of

genes regulating ACh receptor subunits (see above) and increased percentage of NCAM-positive fibres, previously reported also by other studies (Arentson-Lantz et al., 2016; Demangel et al., 2017; Monti et al., 2021). However, we did not find any evidence of an increase in the percentage of fibre type grouping (both for slow and fast fibres) or the presence of flat-shaped/angulated and regenerating neonatal myosin-positive fibres following ULLS. In addition, using iEMG, we observed no spontaneous activity (i.e. no fibrillation potentials of denervated muscle fibres or fasciculation potentials of spontaneously active MUs). Overall these findings suggest initial and partial alterations in muscle innervation status occurring during short-term (10-day) unloading, potentially contributing to the changes in MUP characteristics observed, but in absence of marked denervation/reinnervation processes. Since the propagation velocity of an MFAP increases with increasing muscle fibre diameter (Nandedkar & Stalberg, 1983; Methenitis et al., 2016), fibre diameter variability could also contribute to less synchronous MFAPs propagation (Piasecki et al., 2021). However, this parameter was unchanged after ULLS. Changes in the ion channels dynamics, involved in resting membrane conductance and MFAP propagation along the sarcolemma (Jurkat-Rott & Lehmann-Horn, 2004) and previously reported to be altered with unloading in animal models (Desaphy et al., 2001; Pierno et al., 2002; Tricarico et al., 2010), could represent a possible mechanism underpinning the observed alterations in MUP and NF MUP characteristics. In support of this hypothesis, overall skeletal muscle ion channels gene expression was downregulated following ULLS, highlighting a possible ongoing remodelling of ion channels pool. Since unloading conditions are commonly experienced after injury, surgery and illness, these novel findings of electrophysiological alterations and underlying molecular mechanisms have considerable physiological and clinical relevance.

### Neuromuscular effects of the active recovery period

The last aim of our study was to investigate the effects of an AR intervention based on resistance exercise carried out following ULLS. While it is well established that an AR period can reverse the reduction in muscle mass and function following unloading (Hortobagyi et al., 2000; Suetta et al., 2009; Campbell et al., 2013), whether it could be effective also in counteracting NMJ molecular alterations and changes in electrophysiological properties was unknown. Our results demonstrate that total recovery in muscle function (i.e. MVC) was achieved after 21 days of AR, while after 10 days, muscle strength was not yet restored. Similarly, muscle size, NMJ molecular stability and iEMG outcomes were largely restored within 21 days of AR. CAF concentration at AR21 was no longer significantly different from LS0. Overall, the NMJ and ion channels gene expression seem mostly restored at the end of the AR period. In fact, gene expression at AR21 reached values similar to baseline, with only a limited number of genes that were differentially regulated comparing LS0 and AR21. The strong anti-correlation of GSEA leading-edge genes (Fig. 5) represents an additional robust manifestation of unloading reversal on transcriptomic profile.

Regarding electrophysiological properties, $IDI_{mean}$ (10% MVC only) and MUP complexity were restored at AR21, while NF count and NF MUP duration remained elevated. We also observed a reduction in MUP size (MUP area and duration) at AR21 compared to both LS0 and LS10. Since MUP size is related to the recruitment threshold of a MU (Pope et al., 2016; Del Vecchio et al., 2017), a possible explanation for this finding may be the fibre type shift towards a slower phenotype (increased percentage of type I fibres) observed after the end of AR, which could have led to a higher contribution from low-threshold MUs during the isometric tasks. A second explanation could be that the AR protocol improved overall neuromuscular control during submaximal isometric contractions investigated in this study, so it became more efficient, achieving the target force output with a pool of relatively smaller, lower threshold MUs. Finally, NF MUP jiggle decreased at 10% MVC post-AR. This could suggest that NMJ transmission stability may be improved by resistance training in humans, although it should be considered that smaller MUPs have generally lower NF MUP jiggle. The reasons behind these differences not observed at 25% MVC should be further investigated in future studies.

In conclusion, with this study, we show that human NMJs are destabilized at the molecular level but not functionally impaired by short-term (10 days) unloading, suggesting that NMJ transmission stability is a particularly robust process, at least at relatively low contraction intensities, at which slow-type MUs are recruited. Moreover, our findings highlight that changes in MUP and NF MUP characteristics occur following ULLS, possibly due to alterations in the dynamics of skeletal muscle ion channels, initial signs of axonal damage and partial denervation. Finally, 21 days of resistance training seems sufficient to restore neuromuscular integrity and function to baseline levels.

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

# Additional information

## Data availability statement

RNA-Seq datasets are available at the Gene Expression Omnibus repository: https://www.ncbi.nlm.nih.gov/geo/query/acc.cgi?acc=GSE211204. The other data that support the findings of this study will be made available from the corresponding author upon reasonable request.

## Competing interests

The authors have no conflict of interest to declare.

## Author contributions

F.S., G.V., M.V.F., G.D.V., C.R., and M.V.N. conceptualized and designed the study. M.V.N. obtained the funding. F.S., G.V., G.S. and M.P. performed data collection. F.S. performed iEMG analysis, with the supervision of D.W.S. and J.S.M.P. F.S. analysed *in vivo* muscle morphology and function data. E.M. and S.Z. cut cryosections and carried out immunochemistry and ATPase analysis. G.S. performed ELISA to obtain CAF results. L.M.H. performed RNA extraction, while J.C. and L.G. carried out RNA-Seq analysis. F.S. drafted the manuscript and all the authors revised it. All the authors approved the final version of the manuscript and agreed to be accountable for all aspects of the work in ensuring that questions related to the accuracy or integrity of any part of the work are appropriately investigated and resolved. All persons designated as authors qualify for authorship, and all those who qualify for authorship are listed.

## Funding

The present study was funded by the Italian Space Agency (ASI), MARS-PRE, Project, n. DC-VUM-2017-006. The authors are grateful to ASI for granting these funds to allow all the experiments to be performed.

## Acknowledgements

The authors would like to thank Mr Enrico Roma for the precious suggestions on statistical analysis, Dr Leonardo Nogara and Dr Luana Toniolo for helping during part of the muscle biopsy collection, Dr Tommaso Giacon and Dr Nicola Borasio for helping during iEMG and clinical procedures, Mr Paul Ritsche, Mr Lorenzo Pavan, and Mr Leonardo Cavaggioni for helping during the training sessions and Dr Marco Pirazzini and Dr Giulia Zanetti for the suggestions in the revision of the manuscript.

Open Access Funding provided by Universita degli Studi di Padova within the CRUI-CARE Agreement.

## Keywords

C-terminal agrin fragment, disuse, intramuscular electromyography, jiggle, jitter, motor units

# Supporting information

Additional supporting information can be found online in the Supporting Information section at the end of the HTML view of the article. Supporting information files available:

**Statistical Summary Document**
**Peer Review History**
**Data S1**.
**Data S2**.
**Data S3**.

