## [Peer Review History · The Journal of Physiology]

Effects of short-term unloading and active recovery on human motor unit properties, neuromuscular junction transmission and transcriptomic profile

Fabio Sarto, Dan Stashuk, Martino V Franchi, Elena Monti, Sandra Zampieri, Giacomo Valli, Giuseppe Sirago, Julian Candia, Lisa M Hartnell, Matteo Paganini, Jamie S McPhee, Giuseppe De Vito, Luigi Ferrucci, Carlo Reggiani, and Marco Narici

DOI: 10.1113/JP283381

Corresponding author(s): Fabio Sarto (fabio.sarto.2@phd.unipd.it)

The following individual(s) involved in review of this submission have agreed to reveal their identity: Casper Soendenbroe (Referee #2)

Review Timeline:

Submission Date:	27-May-2022
Editorial Decision:	07-Jul-2022
Revision Received:	29-Jul-2022
Editorial Decision:	08-Aug-2022
Revision Received:	24-Aug-2022
Accepted:	26-Aug-2022

Senior Editor: Scott Powers

Reviewing Editor: Bruno Grassi

Transaction Report:

Dear Mr Sarto,

Re: JP-RP-2022-283381 "Effects of short-term unloading and active recovery on human motor unit properties, neuromuscular junction transmission and transcriptomic profile" by Fabio Sarto, Dan Stashuk, Martino V Franchi, Elena Monti, Sandra Zampieri, Giacomo Valli, Giuseppe Sirago, Julian Candia, Lisa Hartnell, Matteo Paganini, Jamie S McPhee, Giuseppe De Vito, Luigi Ferrucci, Carlo Reggiani, and Marco Narici

Thank you for submitting your manuscript to The Journal of Physiology. It has been assessed by a Reviewing Editor and by 2 expert Referees and I am pleased to tell you that it is considered to be acceptable for publication following satisfactory revision.

The reports are copied at the end of this email. Please address all of the points and incorporate all requested revisions, or explain in your Response to Referees why a change has not been made.

NEW POLICY: In order to improve the transparency of its peer review process The Journal of Physiology publishes online as supporting information the peer review history of all articles accepted for publication. Readers will have access to decision letters, including all Editors' comments and referee reports, for each version of the manuscript and any author responses to peer review comments. Referees can decide whether or not they wish to be named on the peer review history document.

Authors are asked to use The Journal's premium BioRender (<https://biorender.com/>) account to create/redraw their Abstract Figures. Information on how to access The Journal's premium BioRender account is here: <https://physoc.onlinelibrary.wiley.com/journal/14697793/biorender-access> and authors are expected to use this service. This will enable Authors to download high-resolution versions of their figures. The link provided should only be used for the purposes of this submission. Authors will be charged for figures created on this premium BioRender account if they are not related to this manuscript submission.

I hope you will find the comments helpful and have no difficulty returning your revisions within 4 weeks.

Your revised manuscript should be submitted online using the links in Author Tasks Link Not Available.

Any image files uploaded with the previous version are retained on the system. Please ensure you replace or remove all files that have been revised.

REVISION CHECKLIST:

- Article file, including any tables and figure legends, must be in an editable format (eg Word)
- Abstract figure file (see above)
- Statistical Summary Document
- Upload each figure as a separate high quality file
- Upload a full Response to Referees, including a response to any Senior and Reviewing Editor Comments;
- Upload a copy of the manuscript with the changes highlighted.

- A potential 'Cover Art' file for consideration as the Issue's cover image;
- Appropriate Supporting Information (Video, audio or data set https://jp.msubmit.net/cgi-bin/main.plex?form_type=display_requirements#supp).

To create your 'Response to Referees' copy all the reports, including any comments from the Senior and Reviewing Editors, into a Word, or similar, file and respond to each point in colour or CAPITALS and upload this when you submit your revision.

I look forward to receiving your revised submission.

If you have any queries please reply to this email and staff will be happy to assist.

Yours sincerely,

Scott K. Powers
Senior Editor
The Journal of Physiology
<https://jp.msubmit.net>
<http://jp.physoc.org>
The Physiological Society
Hodgkin Huxley House
30 Farringdon Lane
London, EC1R 3AW
UK
<http://www.physoc.org>
<http://journals.physoc.org>

REQUIRED ITEMS:

-Author photo and profile. First (or joint first) authors are asked to provide a short biography (no more than 100 words for one author or 150 words in total for joint first authors) and a portrait photograph. These should be uploaded and clearly labelled with the revised version of the manuscript. See Information for Authors for further details.

-You must start the Methods section with a paragraph headed Ethical Approval. If experiments were conducted on humans confirmation that informed consent was obtained, preferably in writing, that the studies conformed to the standards set by the latest revision of the Declaration of Helsinki, and that the procedures were approved by a properly constituted ethics committee, which should be named, must be included in the article file. If the research study was registered (clause 35 of the Declaration of Helsinki) the registration database should be indicated, otherwise the lack of registration should be noted as an exception (e.g. The study conformed to the standards set by the Declaration of Helsinki, except for registration in a database.). For further information see: <https://physoc.onlinelibrary.wiley.com/hub/human-experiments>

-Please upload separate high-quality figure files via the submission form.

-A Statistical Summary Document, summarising the statistics presented in the manuscript, is required upon revision. It must be on the Journal's template, which can be downloaded from the link in the Statistical Summary Document section here: https://jp.msubmit.net/cgi-bin/main.plex?form_type=display_requirements#statistics

-Papers must comply with the Statistics Policy https://jp.msubmit.net/cgi-bin/main.plex?form_type=display_requirements#statistics

In summary:

-If n {less than or equal to} 30, all data points must be plotted in the figure in a way that reveals their range and distribution. A bar graph with data points overlaid, a box and whisker plot or a violin plot (preferably with data points included) are acceptable formats.

-If $n > 30$, then the entire raw dataset must be made available either as supporting information, or hosted on a not-for-profit repository e.g. FigShare, with access details provided in the manuscript.

-' n ' clearly defined (e.g. x cells from y slices in z animals) in the Methods. Authors should be mindful of pseudoreplication.

-All relevant ' n ' values must be clearly stated in the main text, figures and tables, and the Statistical Summary Document (required upon revision)

-The most appropriate summary statistic (e.g. mean or median and standard deviation) must be used. Standard Error of the Mean (SEM) alone is not permitted.

-Exact p values must be stated. Authors must not use 'greater than' or 'less than'. Exact p values must be stated to three significant figures even when 'no statistical significance' is claimed.

-Statistics Summary Document completed appropriately upon revision

-Please include an Abstract Figure. The Abstract Figure is a piece of artwork designed to give readers an immediate understanding of the research and should summarise the main conclusions. If possible, the image should be easily 'readable' from left to right or top to bottom. It should show the physiological relevance of the manuscript so readers can assess the importance and content of its findings. Abstract Figures should not merely recapitulate other figures in the manuscript. Please try to keep the diagram as simple as possible and without superfluous information that may distract from the main conclusion(s). Abstract Figures must be provided by authors no later than the revised manuscript stage and should be uploaded as a separate file during online submission labelled as File Type 'Abstract Figure'. Please ensure that you include the figure legend in the main article file. All Abstract Figures should be created using BioRender. Authors should use The Journal's premium BioRender account to export high-resolution images. Details on how to use and access the premium account are included as part of this email.

EDITOR COMMENTS

Reviewing Editor:

This is an interesting study asking a central physiological question about how the quality of synaptic transmission to the muscle fibre during voluntary contraction is acutely modified with disuse and resistance training. The study is carried out by a very experienced group of researchers. Both Reviewers liked the study, and stressed, as a positive comment, the broad range of utilized methods.

Reviewer 1 makes a very detailed list of comments identifying several areas of improvement, dealing with data analysis and interpretation. The Reviewer also raises several questions regarding EMG, RNA-seq and statistical analyses. Although a major strength of the submission is the multi-method approach, in some cases data may appear contradictory, and the authors are invited to better focus their data interpretation. Reviewer 2 raises several relatively minor comments related to methods, data reporting an interpretation. Reference to previous similar studies could be improved.

Senior Editor:

Thank you for submitting your work to the Journal of Physiology. Your work has been carefully reviewed by two referees and a review editor. Both reviewers find your work interesting and potentially impactful. Nonetheless, both reviewers have provided numerous suggestions for improving your report prior to making a final decision on publication. We look forward to receiving your revised manuscript.

One of the histograms in the paper does not contain individual data points.

REFEREE COMMENTS

Referee #1:

Overview:

This is an interesting study asking a central physiological question, about how the quality of synaptic transmission to the muscle fibre during voluntary contraction, is acutely modified with disuse and resistance training. The additional supporting measures using molecular, imaging, and anatomical methods provide a rich dataset for investigators to work from.

I think there could be improvements with how the results are focused and interpreted. Likewise, the discussion would benefit from being made concise.

Throughout the results and discussion, it becomes clear that there is a disconnect between the electrophysiology and molecular findings, and part of this may be from how the data were collected, and what that should mean. Because intramuscular EMG recorded MUPs during 10 and 25% of contraction intensity, you recorded from lower threshold MUs that

would be characterized as slow-type, that are more protected in states of MU atrophy, loss and remodeling. Next, muscle biopsies sampled a mixture of MUs, and, the larger-type MUs may have indeed been affected greater by the disuse intervention. Therefore, would the results (NF jitter and jiggle) of this paper only suggest that slower-type, and smaller MUs, would be more resilient during acute disuse?

Specific comments:

1) Introduction, last paragraph. I note the aim of this paper is to study the response of NJM alteration and MUP characteristics. The hypothesis is very broad. Is there a way to make this more precise? Also, are MU firing rates a characteristic of the MUP?

2) Can the methods section be shortened? In many instances you cite prior technical papers.

3) Methods, section 2.5. The term explosiveness is confusing, is there a better descriptor of this measure? Something like rate of force, instead?

4) Methods, section 2.6. Because the needle records EMG activity from the muscle fibre action potential, what is the rationale for insertion so close to the motor point? Does this pose a greater risk of hitting motor twig axons and denervation of the muscle?

4) Methods, section 2.6. Because the intramuscular electrodes were rarely repositioned to a new muscle volume (other than extracting and rotating) is it likely you re-sampled the same MUs? Was this corrected for in the statistical analysis?

5) It would be great to see raw and filtered intramuscular EMG from the VL at the three different study time points (pre, post acute disuse and AR21). Is it possible to add to the manuscript? Would be helpful to illustrate what a MUP looks like in the control, acute disuse, and AR21 state.

6) Methods, section 2.6. From original papers by Stashuk et al., and the numerous papers done by the labs of Stashuk, Doherty and Rice, the threshold for MU inclusion is a minimum of 50 MUPs with a firing rate coefficient of variation (of the mean) <30%. What is the reason that the number of MUPs for MU inclusion has been reduced to 35 MUPs?

7) Methods, section 2.6, paragraph prior to 2.7. I note that voluntary contractions to record MUP characteristics were only at 10 and 25% of MVC. Therefore, are stability measures mainly recorded from low-threshold, slow-type MUs? This is mentioned in one line of the discussion as a limitation, but the results and main discussion don't address this as a likely reason to explain why MUP transmission stability characteristics were unchanged.

8) Methods, section 2.7. Within the same reasoning (point 4, see above), is there greater risk of motor axon injury performing the biopsies so close to the identified motor point?

9) Methods, section 2.7, RNA. Of the RNA samples, what was the RNA integrity score (RIN) average and range?

10) Methods, section 2.7, RNA. I understand that you performed RNA-seq on the entire transcriptome and then applied gene

enrichment guided analysis on the list of differentially expressed genes that passed the EdgeR filter. From the transcriptome, what was the estimated sequencing coverage?

11) Methods, section 2.8. What type of tubes (EDTA, etc.) was the blood collected in?

12) Methods, section 2.9. I understand the EMG measures are from $n = 10$, and muscle biopsies are from $n = 9$. This is clear throughout the abstract and results?

13) Methods, section 2.9. From the list of identified genes, the pathway analysis (Subramanian et al., 2015; and, Jurkat-Rott and Lehmann-Hom, 2004) were originally developed based on microarray methods? Does this pose an issue with using genes identified by RNA-seq (you likely have many more genes identified using RNA-seq)? For example, how many pathways were significantly enriched? And were these 12 extracted pathways relevant for NMJs in the top pathways identified based on statistical significance? For example, would GO enrichment of the same gene list also support your findings?

13) Methods, section 2.9. I may have missed this, but, how many genes were significantly differentially expressed? From this gene list, did you perform a false-discovery rate (FDR) correction prior to pathway enrichment steps?

14) Methods, section 2.9. For the mixed effects linear model of iEMG data, would it also be possible to add a the random variable of muscle insertion location (you performed two per participant?), this could help adjust the model for MU duplications. There are other resampling methods that could also be explored, too.

15) Results, MUP properties. What was the average number of MUPs per participant?

15) Results, MUP properties. In the abstract and introduction, this result is positioned as the main focus of the paper. This section is difficult to read as so many measures are being reported. Also, what measures here support the question you are asking in the introduction? For example MUP area was decreased from acute disuse, but as the reader, I have to work hard to understand this. However, NF MUP area was not observed to have any changes. So what measure should we take as the 'area' result? Does the MUP area change or not? Does MUP area indicate a measure of NMJ stability?

16) Results, MUP properties. In the second last sentence, I remain skeptical that NF jiggle at 10% in the AR21 was changed as both the NF MUP area, and NF MUP jitter were not different. What physiological reasons support that NF jiggle decreases from resistance training? Are there any outliers in your data having a large effect?

17) Table 1, the estimates could benefit from scientific units of measurement.

18) Results, transcriptome. How many genes were differentially expressed for each condition from the number of identified genes that passed the EdgeR filter? Could this be added to the figure legends?

19) Figure 4. The gene expression fold changes in these volcano plots appear to be very small. Would a large functional outcome be the result as the authors speculate in the discussion? Here, it looks like there is a FDR employed with the q-value hash marks, but I don't think this was clearly communicated in the methods.

20) Results, second last paragraph, just above figure 6. Based on the RNA-seq methods used (pooling samples, etc.), I

think the conclusion that gene expression changes from disuse was reversed due to AR21 is a bit of a leap. Also, figure 6 shows that there remained differential gene expression with AR21 (figure 5, panel C).

21) Discussion, first paragraph. 1) Are human NJMs from low-threshold, slow-type MUs already thought to be resilient to the effects of disuse and MU atrophy/ remodeling? 2) Maybe I missed this, but I thought your results showed that MUP characteristics (NF area, jitter and jiggle) were largely unchanged? 3) Neuromuscular changes with regards to what, the gene expression?

22) Discussion, changes in MUP. Can this be shortened and made more relevant to your findings?

23) Discussion, conclusion. Do your results show that the NMJ is destabilized? Based on the results, the EMG data support the the NMJ of low-threshold, slow-type MUs is resilient, and that genes related to the NMJ are modified (could this result be viewed as an acute protective effect, instead?). I remain confused about what MUP characteristics are changed with acute disuse? Do MUP turns and duration really indicate anything physiologically significant at the NMJ? Especially because the MUP NF area, jitter and jiggle were unchanged?

24) Discussion, conclusion. If NF jiggle and jitter were unchanged from acute disuse, was neuromuscular integrity ever compromised? Would muscle fibre changes (figure 7) from higher-threshold, larger-type MUs, have influenced the gene expression changes even more? Would this be a more plausible conclusion?

Referee #2:

This is a very interesting study conducted by a collection of researchers with great expertise in human disuse models. The topic of the study, neuromuscular transmission following disuse and retraining, is highly warranted in the context of patient immobilization. The greatest strength of the study lies in the broad range of methods used to investigate the research question. The relevancy of the study would have been further increased if an old group had also been studied, as disuse likely plays a pivotal role in the development of age-related loss of muscle mass and function.

The authors are relatively open by the fact that some of their findings have previously been reported by others or themselves. The disproportionate loss of force relative to mass (specific force) following disuse has been reported numerous times (Di Girolamo et al., 2021; Marusic et al., 2021), and this serves as the focal point around which one can ponder what mechanism lies beneath. To investigate these mechanism the authors, employ an impressive array of methods, including in vivo measurements, EMG, blood samples, IHC and RNAseq.

The following findings have, to this reviewer's knowledge, previously been reported in relation to disuse:

- TTP63%, reflective of rapid force production, worsen with disuse (Monti et al., 2021).
- Activation capacity decline with disuse (Suetta et al., 2009).
- MUP properties change with disuse (Campbell et al., 2019).
- Muscle morphology change following disuse and retraining (Hvid et al., 2010)
- Number of NCAM+ fibers increase following disuse (Arentson-Lantz et al., 2016; Demangel et al., 2017; Monti et al., 2021)
- CAF increase following disuse (Monti et al., 2021)

The following findings are either completely novel (to this reviewer's knowledge) or oppose previous findings:

- Neurofilament light chain is increased with disuse and might serve as a useful biomarker for nerve damage.
- NMJ dysfunction has been reported following disuse (Grana et al., 1996), but in the present study the authors seen no changes in NMJ stability. Grana et al., also used a 4-week cast immobilization model.
- RNAseq was also used in the groups previous paper, Monti et al., 2021. However, in that paper only a handful of targets were reported. The data in the present study is therefore more comprehensive.
- The percentage of enclosed type II fibers was reduced with retraining, although this was, as pointed out by the authors, likely tied with the change in type II percentage.

To add to this, retraining following disuse is not always investigated, which greatly increases the value of this study. Furthermore, the study overall seems to be well conducted and great expertise within the different methods are found among the list of authors.

There are several points, organized in a section-by-section manner, that I would like the authors to address. I truly hope that the authors can see reason in these comments, and that the comments will increase the value of the work.

Introduction

It should be pointed out that the study by Nishimune et al., mostly refers to animal studies, especially when it comes to pre and post synaptic alterations by exercise: "Physical exercise is well known to maintain NMJ integrity both at its pre- and post-synaptic components, via the action of different neurotrophins (Nishimune et al., 2014)."

Methods

I suggest that the authors consider adding a timeline (LS0, LS10, AR10 and AR21) to increase the readability of the study.

How were the participants recruited and from where? Also, the authors argue that only males were recruited due to increased risk of deep venous thrombosis in females, but surely young and healthy women would be able to complete this protocol safely? Without having checked their references (Bleeker et al., 2004), I would image that health status in general would be more important than sex, when dealing with young individuals.

Please provide the 1RM data, which I assume was measured 3-4 times in total during retraining. If the training data was logged this could also be plotted. This would give an idea of the relative starting point of the legs and how the "disused" leg would catch up.

Regarding ultrasound measurements, why did you decide to average the measurements at 30, 50 and 70% femur length, rather than showing these lengths individually, as you have previously done for the hamstrings (Sarto et al., 2021; Franchi et al., 2022)? Also, how many scans were performed at each site, as if more than one how was this dealt with?

What software was used for DQEMG analysis?

The biopsy method proposed by Aubertin-Leheudre, where the intent is to increase the number of NMJs obtained per biopsy, was used in the present manus. Was this done, because the authors at some point intended to analyze NMJs from humans using this method, but perhaps did not get enough biopsies with NMJs? The rationale for choosing this method should at least be clear.

Also, in the original publication, Aubertin-Leheudre et al., report to get ~40 NMJs per biopsy as opposed to ~0 when using normal method (Aubertin-Leheudre et al., 2020). The authors in the present study argue, in relation to the RNAseq and IHC data, that these genes and proteins that they measure are upregulated extra-synaptically when the muscle fiber is denervation/destabilized. This I would normally agree with, but since the special biopsy technique was used, this claim is problematic, as some of the "positive" signal could in fact come from actual NMJs in the biopsies? Please consider this in your discussion AND quantify NMJs in your biopsy sections using α -bungarotoxin and/or nestin (no graph needed, just the number of NMJs per biopsy) (Soendenbroe et al., 2022).

Finally, there is some ambiguity when it comes to biopsy sampling. Biopsies were sampled at three time points, but it is not clear whether the order was randomized or fixed, and how close were the sample sites were to one another? The authors also mention that VL usually have three distinct motor points, but they are spatially separated, with some of them located at a place where biopsies would not be taken from (Gobbo et al., 2014). So, assuming that biopsies at all time points were taken ~2 cm from the same motor point, the distance between sites might be too little avoid effects of pre-sampling (Vissing et al., 2005). And perhaps this could also influence on iEMG measurements, since they too were performed close to the motor point. All these elements have to be considered and specified in the text, and some sort of overview figure showing exactly how these things were performed on each thigh, would be greatly appreciated.

Add RRID for NCL-MHCd. AB_563901 I think?

I appreciate the effort to run positive controls, but I do not understand, from your description, how an antibody made in mouse (NCL-MHCd) was used on mouse tissue? Please specify.

What objective (and numerical aperture) was used?

How was imaging performed in terms of exposure times etc., fixed or variable conditions?

These sentences say the same thing: "Negative controls were performed by omitting the primary antibodies from sample incubations." AND "...while negative controls were performed by omitting the primary antibodies from sample incubations"

Were biopsy analyses (IHC, morphology) performed blinded for leg/time point?

The following description of enclosed fibers is not entirely clear to me, considering the results are shown as % grouped fibers out of all fibers: "For the quantification of large and very large fibre type clusters (>10 or >20 fibres, respectively), the method of "enclosed fibre"". Also, please be consistent in the use of "grouped" versus "enclosed", as the former seems to refer to fibers of the same type that are connected to each other, while the latter refer to a fiber that is complete surround by fibers of its own type.

Also, fiber diameter variability was also assessed. However, another measurement, usually annotated "shape factor", has previously been used to analyze variability in muscle morphology (Barnouin et al., 2017; Messa et al., 2020). Did the authors consider using this measurement?

In relation to the RNAseq, I appreciate that it has been performed by a third-party company, which is okay. But there are central parts of the method section, which is simply too unspecific, for example in relations to products used (cat. number, amount, etc.). Please specify as much as possible.

Importantly, as this reviewer understands it, it is only genes that are "known" to be involved in NMJ and skeletal muscle ion channel regulation, that is shown in the data sets? In other words, a fraction of all the genes that were detected was chosen

for this analysis. While it is not in my place to decide whether this approach is correct, I believe that this should be stated in VERY clear terms in both the methods, results and discussion sections and in corresponding figure legends. And it would be of value to say something about how large/small this fraction was?

Please elaborate and specify the GSEA approach.

Please provide references for neurofilament light chain to be a "well-established" marker of axonal damage. Again, the level of detail on this method is very low (company analysis), so please specify as much as possible.

Why was CAF used for power calculations, and not a more classic measurement, like MVC or mass?

In relation to the statistical analyses of the EMG data, it seems that the authors have used a common approach with this type of data, which is use an average of ALL motor unit measurements pooled together, and not per subject, which greatly increases the n. They use a generalized linear mixed effect model (fixed effect: time; cluster variable: subject) and provide a reference in an attempt to justify this (Yu et al., 2022). However, it does not seem appropriate to treat a single MU as a statistical unit, instead it should be a single individual. If this was done in a biopsy analysis, n's of several thousands would not be uncommon when evaluated changes in for instance fiber size. I suggest that the authors 1) consider these elements and their limitations in their discussion of the data, 2) specify in the figure legends exactly how statistical testing was performed, and 3) provide complimentary figures and statistical outcomes of these data sets (EMG measurements) performed with subject as the statistical unit. This last element can figure in the response letter only, if desired.

Results / figures

Please write statistical test in figure legend, especially since regular or mixed effects were used depending on missing values.

"Interestingly, for both muscles, CSA mean at AR21 was higher than LS0 and AR21...". Both muscles here refers to vastus lateralis and quadriceps, but the latter is not a muscle. Also, it seems it should be LS10 and not AR21 in that sentence?

Figure 2+3: Show individual subject values. Also, could the authors provide some representative curves for both MUP, NF and NMJ stability. Please also specify within curves what each measurement refers to. See these papers for examples (Piasecki et al., 2021; Guo et al., 2021).

Muscle morphology. Please also provide fiber CSA instead of only ferret diameter. Representative images of ATPase staining would be greatly appreciated.

Please show the NCAM and MyHCn data. Use individual values and median, given the lack of normal distribution. Please also provide representative images.

Figure 4+5+6. Please name all significant genes or provide a corresponding list of significant genes. Right now the majority of information is lost as many genes remain unspecified.

Figure 5: It is not clear among the clustered targets in the middle which gene name belongs to which dot. Consider using a small line between name and dot. Also, please check whether text and figure legend are aligned in terms of what is down- and upregulation in the four fields.

Figure 7: The significant increase in CAF (5,4 %, $p=0.038$) seems very slim. Please provide the individual data points used in the statistical analysis in the response. Also, E and F should have the same y-axis.

Discussion

Please discuss why only young were studied, as an old group in my view would greatly enhance the relevancy of the work?

As the authors note themselves, there is a very dramatic reduction in type II fiber percentage (corresponding increase in type I). Since ATPase was performed to only distinguish between type I and II, it is not clear whether this results from isoform specific changes. Is the dramatic change related to altered coexpression of isoforms or perhaps a reduction in type IIx? (Andersen et al., 1999 p.19; Andersen & Aagaard, 2000).

The percentage of enclosed type II fibers seems very high at baseline (LS0), considering this is young healthy men. Messa et al., 2020, reported, using the same method by Jennekens et al., 1971, a percentage of type II enclosed fibers of ~2% in young male ($n=14$) and female ($n=8$) controls (Jennekens et al., 1971; Messa et al., 2020). Please discuss.

In relation to NMJ stability, the authors base this only on NF jitter and jiggle, but would the other NF EMG measurements not also inform on this? Please explain.

"Similarly, muscle size, NMJ stability, MUP and NF MUP characteristics were restored within 21 days of AR.". It does not make sense that it has just been concluded that NMJ stability was not affected by disuse, yet now it is restored?

"Overall, the AR period counteracted the effects of unloading on the NMJ and ion channel gene expression.". Since there was no non-exercising control group, it can in fact no be concluded that it was AR that counteracted the effects of unloading, it could also be time itself. This point should perhaps be brought forward by the authors, and perhaps they know of other studies that have compared active and passive recovery following disuse. Suetta et al., 2004 compared standard recovery with strength training + standard recovery in elderly men and women following hip-replacement surgery and found that exercise was crucial in order to recover strength (Suetta et al., 2004). But then again, this was elderly people, following surgery, presumably in pain and therefore perhaps not relevant to this study.

References

- Andersen JL & Aagaard P (2000). Myosin heavy chain IIX overshoot in human skeletal muscle. *Muscle & Nerve* 23, 1095-1104.
- Andersen JL, Terzis G & Kryger A (1999). Increase in the degree of coexpression of myosin heavy chain isoforms in skeletal muscle fibers of the very old. *Muscle Nerve* 22, 449-454.
- Arentson-Lantz EJ, English KL, Paddon-Jones D & Fry CS (2016). Fourteen days of bed rest induces a decline in satellite cell content and robust atrophy of skeletal muscle fibers in middle-aged adults. *J Appl Physiol* (1985) 120, 965-975.
- Aubertin-Leheudre M, Pion CH, Vallée J, Marchand S, Morais JA, Bélanger M & Robitaille R (2020). Improved Human Muscle Biopsy Method To Study Neuromuscular Junction Structure and Functions with Aging. *J Gerontol A Biol Sci Med Sci* 75, 2098-2102.
- Barnouin Y, McPhee JS, Butler-Browne G, Bosutti A, De Vito G, Jones DA, Narici M, Behin A, Hogrel J-Y & Degens H (2017). Coupling between skeletal muscle fiber size and capillarization is maintained during healthy aging. *J Cachexia Sarcopenia Muscle* 8, 647-659.
- Bleeker MWP, Hopman MTE, Rongen GA & Smits P (2004). Unilateral lower limb suspension can cause deep venous thrombosis. *Am J Physiol Regul Integr Comp Physiol* 286, R1176-1177.

- Campbell M, Varley-Campbell J, Fulford J, Taylor B, Mileva KN & Bowtell JL (2019). Effect of Immobilisation on Neuromuscular Function In Vivo in Humans: A Systematic Review. *Sports Med* 49, 931-950.
- Demangel R, Treffel L, Py G, Brioché T, Pagano AF, Bareille M-P, Beck A, Pessemeesse L, Candau R, Gharib C, Chopard A & Millet C (2017). Early structural and functional signature of 3-day human skeletal muscle disuse using the dry immersion model. *J Physiol* 595, 4301-4315.
- Di Girolamo FG, Fiotti N, Milanović Z, Situlin R, Mearelli F, Vinci P, Šimunič B, Pišot R, Narici M & Biolo G (2021). The Aging Muscle in Experimental Bed Rest: A Systematic Review and Meta-Analysis. *Front Nutr*; DOI: 10.3389/fnut.2021.633987.
- Franchi MV, Sarto F, Simunič B, Pišot R & Narici MV (2022). Early Changes of Hamstrings Morphology and Contractile Properties During 10 Days of Complete Inactivity. *Med Sci Sports Exerc*; DOI: 10.1249/MSS.0000000000002922.
- Gobbo M, Maffiuletti NA, Orizio C & Minetto MA (2014). Muscle motor point identification is essential for optimizing neuromuscular electrical stimulation use. *J Neuroeng Rehabil* 11, 17.
- Grana EA, Chiou-Tan F & Jaweed MM (1996). Endplate dysfunction in healthy muscle following a period of disuse. *Muscle & Nerve* 19, 989-993.
- Guo Y, Piasecki J, Swiecicka A, Ireland A, Phillips BE, Atherton PJ, Stashuk D, Rutter MK, McPhee JS & Piasecki M (2021). Circulating testosterone and dehydroepiandrosterone are associated with individual motor unit features in untrained and highly active older men. *Geroscience*; DOI: 10.1007/s11357-021-00482-3.
- Hvid L, Aagaard P, Justesen L, Bayer ML, Andersen JL, Ørtenblad N, Kjaer M & Suetta C (2010). Effects of aging on muscle mechanical function and muscle fiber morphology during short-term immobilization and subsequent retraining. *J Appl Physiol* (1985) 109, 1628-1634.
- Jennekens FG, Tomlinson BE & Walton JN (1971). Data on the distribution of fibre types in five human limb muscles. An autopsy study. *J Neurol Sci* 14, 245-257.
- Marusic U, Narici M, Simunic B, Pisot R & Ritzmann R (2021). Nonuniform loss of muscle strength and atrophy during bed rest: a systematic review. *J Appl Physiol* (1985) 131, 194-206.
- Messa GAM, Piasecki M, Rittweger J, McPhee JS, Koltai E, Radak Z, Simunic B, Heinonen A, Suominen H, Korhonen MT & Degens H (2020). Absence of an aging-related increase in fiber type grouping in athletes and non-athletes. *Scand J Med Sci Sports* 30, 2057-2069.
- Monti E, Reggiani C, Franchi MV, Toniolo L, Sandri M, Armani A, Zampieri S, Giacomello E, Sarto F, Sirago G, Murgia M, Nogara L, Marcucci L, Ciciliot S, Šimunic B, Pišot R & Narici MV (2021). Neuromuscular junction instability and altered intracellular calcium handling as early determinants of force loss during unloading in humans. *J Physiol* 599, 3037-3061.
- Nishimune H, Stanford JA & Mori Y (2014). ROLE of exercise in maintaining the integrity of the neuromuscular junction: Invited Review: Exercise and NMJ. *Muscle & Nerve* 49, 315-324.
- Piasecki J, Inns TB, Bass JJ, Scott R, Stashuk DW, Phillips BE, Atherton PJ & Piasecki M (2021). Influence of sex on the age-related adaptations of neuromuscular function and motor unit properties in elite masters athletes. *J Physiol* 599, 193-205.
- Sarto F, Spörri J, Fitze DP, Quinlan JI, Narici MV & Franchi MV (2021). Implementing Ultrasound Imaging for the Assessment of Muscle and Tendon Properties in Elite Sports: Practical Aspects, Methodological Considerations and Future Directions. *Sports Med* 51, 1151-1170.
- Soendenbroe C, Flindt Heisterberg MF, Schjerling P, Kjaer M, Andersen JL & Mackey AL (2022). Human skeletal muscle acetylcholine receptor gene expression in elderly males performing heavy resistance exercise. *Am J Physiol Cell Physiol*; DOI: 10.1152/ajpcell.00365.2021.
- Suetta C, Aagaard P, Rosted A, Jakobsen AK, Duus B, Kjaer M & Magnusson SP (2004). Training-induced changes in muscle CSA, muscle strength, EMG, and rate of force development in elderly subjects after long-term unilateral disuse. *J Appl Physiol* (1985) 97, 1954-1961.
- Suetta C, Hvid LG, Justesen L, Christensen U, Neergaard K, Simonsen L, Ortenblad N, Magnusson SP, Kjaer M & Aagaard P (2009). Effects of aging on human skeletal muscle after immobilization and retraining. *J Appl Physiol* (1985) 107, 1172-1180.
- Vissing K, Andersen JL & Schjerling P (2005). Are exercise-induced genes induced by exercise? *FASEB J* 19, 94-96.
- Yu Z, Guindani M, Grieco SF, Chen L, Holmes TC & Xu X (2022). Beyond t test and ANOVA: applications of mixed-effects models for more rigorous statistical analysis in neuroscience research. *Neuron* 110, 21-35.

END OF COMMENTS

Confidential Review

27-May-2022

EDITOR COMMENTS

Reviewing Editor:

This is an interesting study asking a central physiological question about how the quality of synaptic transmission to the muscle fibre during voluntary contraction is acutely modified with disuse and resistance training. The study is carried out by a very experienced group of researchers. Both Reviewers liked the study, and stressed, as a positive comment, the broad range of utilized methods.

Reviewer 1 makes a very detailed list of comments identifying several areas of improvement, dealing with data analysis and interpretation. The Reviewer also raises several questions regarding EMG, RNA-seq and statistical analyses. Although a major strength of the submission is the multi-method approach, in some cases data may appear contradictory, and the authors are invited to better focus their data interpretation. Reviewer 2 raises several relatively minor comments related to methods, data reporting and interpretation. Reference to previous similar studies could be improved.

We are grateful to the Reviewing Editor for thoroughly assessing our manuscript and for identifying some areas for improvement together with the two expert Reviewers. We have tried to take into consideration the Reviewers' suggestions as best as we could, in order to amend and improve the manuscript accordingly.

Senior Editor:

Thank you for submitting your work to the Journal of Physiology. Your work has been carefully reviewed by two referees and a review editor. Both reviewers find your work interesting and potentially impactful. Nonetheless, both reviewers have provided numerous suggestions for improving your report prior to making a final decision on publication. We look forward to receiving your revised manuscript.

One of the histograms in the paper does not contain individual data points.

We are grateful to the Senior Editor for coordinating the review process.

Please find our point-by-point response. Our comments/feedback are shown in blue font. The corrections applied to the new version of the manuscript will also be displayed in blue font.

To exhaustively answer the Reviewers, we provided some additional figures and tables that are named with letters (Figure A, Figure B etc.) instead of numbers to avoid confusion with the figures and tables presented in the manuscript.

REFEREES' COMMENTS

Referee #1:

Overview:

This is an interesting study asking a central physiological question, about how the quality of synaptic transmission to the muscle fibre during voluntary contraction, is acutely modified with disuse and resistance training. The additional supporting measures using molecular, imaging, and anatomical methods provide a rich dataset for investigators to work from.

I think there could be improvements with how the results are focused and interpreted. Likewise, the discussion would benefit from being made concise.

We thank the Reviewer for the positive feedback regarding our work and for detecting some areas for improvement.

Throughout the results and discussion, it becomes clear that there is a disconnect between the electrophysiology and molecular findings, and part of this may be from how the data were collected, and what that should mean. Because intramuscular EMG recorded MUPs during 10 and 25% of contraction intensity, you recorded from lower threshold MUs that would be characterized as slow-type, that are more protected in states of MU atrophy, loss and remodeling. Next, muscle biopsies sampled a mixture of MUs, and, the larger-type MUs may have indeed been affected greater by the disuse intervention. Therefore, would the results (NF jitter and jiggle) of this paper only suggest that slower-type, and smaller MUs, would be more resilient during acute disuse?

We thank the Reviewer for the observation. As stated in the Discussion and Conclusions, we acknowledge that our findings might be limited to relatively low-intensity contractions.

However, data on MUs adaptations with disuse is very limited and conflicting. One study (Duchateau & Hainaut, 1990) reported that firing rate modulation of hand muscles is more affected in low- than in high-threshold MUs, while another study reported opposite results (Seki *et al.*, 2001). Further evidence may be obtained from quantitative (fibre size) and qualitative (mechanical properties) fibre changes in response to disuse. The only two studies that performed this kind of analysis employing the ULLS model showed that (i) decreases in peak force (P_o) were detectable only in type I fibre for both soleus and gastrocnemius after 12-day ULLS (Widrick *et al.*, 2002) and (ii) that similar CSA and P_o /CSA decreases of fast and slow fibres were observed with 3-week ULLS in the vastus lateralis (Brocca *et al.*, 2015). With other models of disuse/microgravity (bed rest, lower limb immobilization, dry immersion, spaceflight) findings are conflicting (Widrick *et al.*, 2001; Fitts *et al.*, 2010; Hvid *et al.*, 2011; Brocca *et al.*, 2012; Demangel *et al.*, 2017; Monti *et al.*, 2021), but most of these point towards a greater impairment (greater decreases in fibre size and force) of type I fibre (Widrick *et al.*, 2001; Fitts *et al.*, 2010; Demangel *et al.*, 2017; Monti *et al.*, 2021).

Overall, we conclude that there is no evidence that low-threshold, slow-type MUs could potentially be more resilient in response to disuse. Conversely, the present findings may suggest the opposite (greater impairment of slow-type MUs), although more evidence is needed to confirm this hypothesis.

Nevertheless, we tried to give more emphasis to the Reviewer's suggestion by adding the following sentence in the Discussion:

"Indeed, we recorded the activity only of low-threshold, slow-type MUs and these may have been differently affected by unloading compared to high-threshold, fast-type MUs."

Specific comments:

1) Introduction, last paragraph. I note the aim of this paper is to study the response of NJM alteration and MUP characteristics. The hypothesis is very broad. Is there a way to make this more precise? Also, are MU firing rates a characteristic of the MUP?

We thank the Reviewer for the suggestion; we tried to make our hypothesis more specific. Now it reads:

"Our hypotheses were that (i) 10-day ULLS, would be sufficient to cause NMJ molecular alterations, possibly resulting in NMJ transmission stability impairment, and increase MUP complexity and decrease MUs firing rate, in association with loss of muscle function and (ii) 21-day AR would counteract these neuromuscular changes."

Firing rates are usually considered a characteristic of motoneurons, not of the MUP (motor units potential), so we preferred to keep them separated.

2) Can the methods section be shortened? In many instances you cite prior technical papers.

We tried to delete several sentences that were not strictly necessary from the methods section to shorten it. However, it should be stated that we were asked to add several technical details in the methods section by Reviewer 2.

3) Methods, section 2.5. The term explosiveness is confusing, is there a better descriptor of this measure? Something like rate of force, instead?

We thank the Reviewer for the suggestion. We amended it as "the capacity for rapid force production".

4) Methods, section 2.6. Because the needle records EMG activity from the muscle fibre action potential, what is the rationale for insertion so close to the motor point? Does this pose a greater risk of hitting motor twig axons and denervation of the muscle?

We thank the Reviewer for the observation. Our iEMG methods were based on previous studies (Piasecki *et al.*, 2016b, 2016a; Jones *et al.*, 2021). To the best of our knowledge, no particular side effects were reported in their studies (our co-author JSMP was the principal investigator of these previous works), despite the considerable sample size. Thus, we believe that this approach does not represent a greater risk.

4) Methods, section 2.6. Because the intramuscular electrodes were rarely repositioned to a new muscle volume (other than extracting and rotating) is it likely you re-sampled the same MUs? Was this corrected for in the statistical analysis?

We agree with the Reviewer that, despite a consistent and deliberate protocol for repositioning the needle detection surface between muscle contractions, resampling of the same MUs is possible with our approach. However, it should be considered that the area of the detection surface of the needles used is only 0.07 mm², thus slight movements of the needle should be sufficient to sample different information. As such, even when MUs were sometimes re-sampled, we likely collected our data from different sets of fibres and NMJs of the same MUs.

5) It would be great to see raw and filtered intramuscular EMG from the VL at the three different study time points (pre, post acute discuse and AR21). Is it possible to add to the manuscript? Would be helpful to illustrate what a MUP looks like in the control, acute disuse, and AR21 state.

We thank the Reviewer for the suggestion. Representative MUP and NF MUP waveforms recorded during the three different time points have been added to Figure 2 and Figure 3 of the manuscript.

6) Methods, section 2.6. From original papers by Stashuk et al., and the numerous papers done by the labs of Stashuk, Doherty and Rice, the threshold for MU inclusion is a minimum of 50 MUPs with a firing rate coefficient of variation (of the mean) <30%. What is the reason that the number of MUPs for MU inclusion has been reduced to 35 MUPs?

We thank the Reviewer for the observation. Previous studies that used 50 MUPs as a threshold for MUP train inclusion were evaluating motor unit number estimate (MUNE) values obtained via decomposition-enhanced spike-triggered averaging and therefore required additional MUPs within a train to reduce noise in surface MUP estimates. We believe that 35 isolated MUPs provide statistical confidence, as our NF MUP raster is entirely composed of isolated (i.e. not contaminated by other MUs) MUPs. Furthermore, it should be considered that in the present work we employed a later version of the DQEMG software which less often merges MUPTs, thus other criteria, such as a firing rate coefficient of variation (of the mean) <30%, were not necessary.

7) Methods, section 2.6, paragraph prior to 2.7. I note that voluntary contractions to record MUP characteristics were only at 10 and 25% of MVC. Therefore, are stability measures mainly recorded from low-threshold, slow-type MUs? This is mentioned in one line of the discussion as a limitation, but the results and main discussion don't address this as a likely reason to explain why MUP transmission stability characteristics were unchanged.

For a detailed discussion of these aspects, we refer the Reviewer to our first answer in the "Overview" (before "Specific comments").

8) Methods, section 2.7. Within the same reasoning (point 4, see above), is there greater risk of motor axon injury performing the biopsies so close to the identified motor point?

In our laboratory, so far we collected ~85 vastus lateralis biopsies following this approach both in young and older adults, with no side effects reported. We conclude that sampling at 2 cm from the motor point is safe and does not represent a greater risk.

9) Methods, section 2.7, RNA. Of the RNA samples, what was the RNA integrity score (RIN) average and range?

The RNA integrity score average was 9.21 (range: 7.90-9.86). We added this information to Methods section 2.7.

10) Methods, section 2.7, RNA. I understand that you performed RNA-seq on the entire transcriptome and then applied gene enrichment guided analysis on the list of differentially expressed genes that passed the EdgeR filter. From the transcriptome, what was the estimated sequencing coverage?

As explained in the Methods section 2.7, samples have 295 to 419 million pass filter reads (average = 367 million reads per sample), which is significantly above the recommended coverage for Illumina RNA-Seq experiments. Notice that coverage in RNA-Seq experiments is typically expressed as the number of reads, and not as a ratio of sequenced bases divided by the number of bases in the reference (which is what is typically used in DNA-Seq experiments).

11) Methods, section 2.8. What type of tubes (EDTA, etc.) was the blood collected in?

The blood was collected in Gel Clot Activator tubes (368969, BD Diagnostic, Oxford UK) in order to separate the serum from the other blood components.

12) Methods, section 2.9. I understand the EMG measures are from n = 10, and muscle biopsies are from n = 9. This is clear throughout the abstract and results?

We thank the Reviewer for allowing us to clarify this point. For iEMG measures n=10 just at LS10; while for biopsies outcomes n=9 only at AR21. In all the other time points n=11. Thus, we employed statistical models (generalized linear mixed models and mixed-effects ANOVA for iEMG measures and biopsies outcomes, respectively) that can handle missing values. Therefore, results are always referred to all the 11 participants. All these details are now reported in the Statistical Summary Document (Supporting Information section).

13) Methods, section 2.9. From the list of identified genes, the pathway analysis (Subramanian et al., 2015; and, Jurkat-Rott and Lehmann-Hom, 2004) were originally developed based on microarray methods? Does this pose an issue with using genes identified by RNA-seq (you likely have many more genes identified using RNA-seq)? For example, how many pathways were significantly enriched? And were these 12 extracted pathways relevant for NMJs in the top pathways identified based on statistical significance? For example, would GO enrichment of the same gene list also support your findings?

GSEA as implemented in the R package fgsea (Korotkevich *et al.*, 2021) is a standard tool of analysis to identify enriched pathways from RNA-Seq data. Since the scope of this work is hypothesis-driven (as opposed to an exploratory, fully data-driven approach) we did not expand the pathway characterization to all other pathways available (e.g from the Molecular Signatures database, including Gene Ontology). Here, GSEA was primarily used to extract NMJ-associated leading-edge genes (Figure 5), which show, in turn, that changes between baseline and 10 days of ULLS are essentially undone upon retraining.

13) Methods, section 2.9. I may have missed this, but, how many genes were significantly differentially expressed? From this gene list, did you perform a false-discovery rate (FDR) correction prior to pathway enrichment steps?

Our Gene Set Enrichment Analysis procedure does not use a hard cut-off to select differentially expressed genes, therefore FDR correction prior to pathway enrichment is not needed. We provide additional details on the GSEA procedure in Methods section 2.7.

14) Methods, section 2.9. For the mixed effects linear model of iEMG data, would it also be possible to add a the random variable of muscle insertion location (you performed two per participant?), this could help adjust the model for MU duplications. There are other resampling methods that could also be explored, too.

Only one insertion was performed at each time point. Moreover, as explained above, even in the case we re-sampled some MUs, the shape of MUPs included will be different, as collected from different fibres and NMJs from that MU. Thus we do not believe it could be considered entirely a duplication.

15) Results, MUP properties. What was the average number of MUPs per participant?

We thank the Reviewer for allowing us to clarify this point:

- for MUPs the average per participant at each time point was 24.77 (12.8) at 10% and 36.73 (16.0) at 25%
- for NF MUPs the average per participant at each time point was 15.0 (10.26) at 10% and 21.5 (11.96) at 25%

This information has been added to the manuscript.

15) Results, MUP properties. In the abstract and introduction, this result is positioned as the main focus of the paper. This section is difficult to read as so many measures are being reported. Also, what measures here support the question you are asking in the introduction? For example MUP area was decreased from acute disuse, but as the reader, I have to work hard to understand this. However, NF MUP area was not observed to have any changes. So what measure should we take as the 'area' result? Does the MUP area change or not? Does MUP area indicate a measure of NMJ stability?

We reworded some sentences in the Results section titled "MUP properties and NMJ transmission stability". We hope this section reads more clearly now.

MUP area represents the size of the MU and reflects the number and sizes of the MU fibres contributing to the MUP. While, NF MUP area can be considered a measure of fibre density (i.e. how many fibre are detected in a particular area, in close proximity, to the needle electrode, within ~350 μm). Since changes in NF MUP area can be difficult to interpret and in order to make the manuscript easier to understand, we decided to remove this parameter from the manuscript.

16) Results, MUP properties. In the second last sentence, I remain skeptical that NF jiggle at 10% in the AR21 was changed as both the NF MUP area, and NF MUP jitter were not different. What physiological reasons support that NF jiggle decreases from resistance training? Are there any outliers in your data having a large effect?

We thank the Reviewer for pointing this out. We did not expect this mild decrease in NF Jiggle with AR21 at 10% MVC either. However, we were not able to detect any outlier at AR21 (see Figure A). When removing the only clear outlier detected at LS0, the results were unchanged.

From a physiological perspective, in animal models, exercise training is known to increase quantal content, the amount of neurotransmitter released with a single stimulus, safety factor and alter the spontaneous release of neurotransmitter (Deschenes, 2019). But whether this can lead to an improved NMJ transmission stability remains unknown in humans. Hence we agree that further studies are required to establish whether NMJ transmission can be improved by resistance training.

Figure A: Distribution of the NF MUP Jiggle data at 10% MVC. Histograms (A) and (violin box plots (B) at each time point. Each dots represent a motor unit. LS0: baseline; LS10: after 10 days of unilateral lower limb suspension; AR21: after 3 weeks of active recovery

17) Table 1, the estimates could benefit from scientific units of measurement.

Scientific units of all the parameters have been added.

18) Results, transcriptome. How many genes were differentially expressed for each condition from the number of identified genes that passed the EdgeR filter? Could this be added to the figure legends?

We added details on the number of genes analysed in the main text (“NMJ and ion channels transcriptomic profile” subsection) and captions of Figures 4 and 6.

19) Figure 4. The gene expression fold changes in these volcano plots appear to be very small. Would a large functional outcome be the result as the authors speculate in the discussion? Here, it looks like there is a FDR employed with the q-value hash marks, but I don't think this was clearly communicated in the methods.

We clarified the use of FDR correction and q-value reference lines (main text, “NMJ and ion channels transcriptomic profile” subsection). Despite relatively modest gene expression fold changes, very significant p-values result from the paired nature of our dataset. Therefore, consistent changes in

the same direction for most/all subjects in the cohort yield robust results potentially indicative of functional outcomes.

20) Results, second last paragraph, just above figure 6. Based on the RNA-seq methods used (pooling samples, etc.), I think the conclusion that gene expression changes from disuse was reversed due to AR21 is a bit of a leap. Also, figure 6 shows that there remained differential gene expression with AR21 (figure 5, panel C).

Please note that this comment makes reference to “Figure 5, panel C” but our Figure 5 does not have multiple panels. To clarify, our claim that “overall, these findings suggest that genes of NMJ and skeletal muscle ion channels underwent adaptive changes between LS0 and LS10, which are mostly reversed at AR21” is collectively supported by the results presented in Figures 4, 5, and 6. On the one hand, PCA plots (Figure 4A and Figure 6A) show that LS0 and AR21 datapoints form nearby and partially overlapping clusters, whereas LS10 datapoints form a clearly distinct, separate cluster. On the other hand, volcano plots showing comparisons between LS10 and LS0 (Figure 4B and Figure 6B) as well as between LS10 and AR21 (Figure 4C and Figure 6C) show much stronger differences than those evidenced between LS0 and AR21 (Figure 4D and Figure 6D). Finally, the strong anti-correlation of leading-edge genes shown in Figure 5 is, in our view, a robust manifestation of disuse reversal.

21) Discussion, first paragraph. 1) Are human NJMs from low-threshold, slow-type MUs already thought to be resilient to the effects of disuse and MU atrophy/ remodeling? 2) Maybe I missed this, but I thought your results showed that MUP characteristics (NF area, jitter and jiggle) were largely unchanged? 3) Neuromuscular changes with regards to what, the gene expression?

We thank the Reviewer for offering these points of discussion:

1) Briefly, there is no evidence that low-threshold, slow-type MUs are less affected by disuse. For a detailed discussion of these aspects, we refer the Reviewer to our first answer in the “Overview” (before “Specific comments”).

2) The MUP properties that were affected by ULLS are: MUP turns, IDImean, NF MUP duration and NF count, overall suggesting increased MUP and NF MUP complexity and decreased motor unit firing rate.

3) “Neuromuscular changes” was used as an umbrella term with regards to alterations in muscle function parameters (MVC, TTP63%, AC) and MU properties (MUP complexity, firing rate).

22) Discussion, changes in MUP. Can this be shortened and made more relevant to your findings?

We thank the Reviewer for the suggestion. After careful evaluation, we decided not to shorten the Discussion section titled “Changes in MUP characteristics with unloading and underlying mechanisms”. We believe that ~600 words are appropriate for the central part of the Discussion and we feel that if shortened this section would not be complete.

23) Discussion, conclusion. Do your results show that the NMJ is destabilized? Based on the results, the EMG data support the the NMJ of low-threshold, slow-type MUs is resilient, and that genes related to the NMJ are modified (could this result be viewed as an acute protective effect, instead?). I remain confused about what MUP characteristics are changed with acute disuse? Do MUP turns

and duration really indicate anything physiologically significant at the NMJ? Especially because the MUP NF area, jitter and jiggle were unchanged?

As explained in the manuscript, we believe that from a molecular perspective, human NMJ is destabilized by short-term unloading. CAF increased, as in our previous bed rest paper (Monti *et al.*, 2021), and this was accompanied by changes in the expression of several NMJ genes. We agree with the Reviewer that upregulation of some of them could point towards a compensatory response (e.g. *GDNF*, *WNT4*), but downregulation of many other genes seems clearly indicative of NMJ damage/instability (e.g. *COLQ*, *HOMER2*, *PPARGC1A*, *NT4*). Overall, independently of the direction of the changes, we believe that this is a sign of altered NMJ transcriptome response with unloading. However, from a functional perspective, as pointed out by the Reviewer, NMJ transmission stability of low-threshold, slow-type MUs seems resilient.

The MUP properties that were changed with disuse are MUP turns, IDImean, NF MUP duration and NF count. We do not think that changes in these properties are indicative of anything physiologically significant at the NMJ, but are more likely related to initial axonal damage, partial denervation and altered skeletal muscle ion channel dynamics, as explained in the manuscript.

24) Discussion, conclusion. If NF jiggle and jitter were unchanged from acute disuse, was neuromuscular integrity ever compromised? Would muscle fibre changes (figure 7) from higher-threshold, larger-type MUs, have influenced the gene expression changes even more? Would this be a more plausible conclusion?

“Neuromuscular integrity and function” is a general term that includes also the muscle function parameters (MVC, TTP63%, AC) and MU properties (MUP complexity, firing rate) that were instead clearly affected by unloading.

Referee #2:

This is a very interesting study conducted by a collection of researchers with great expertise in human disuse models. The topic of the study, neuromuscular transmission following disuse and retraining, is highly warranted in the context of patient immobilization. The greatest strength of the study lies in the broad range of methods used to investigate the research question. The relevancy of the study would have been further increased if an old group had also been studied, as disuse likely plays a pivotal role in the development of age-related loss of muscle mass and function.

The authors are relatively open by the fact that some of their findings have previously been reported by others or themselves. The disproportionate loss of force relative to mass (specific force) following disuse has been reported numerous times (Di Girolamo *et al.*, 2021; Marusic *et al.*, 2021), and this serves as the focal point around which one can ponder what mechanism lies beneath. To investigate these mechanism the authors, employ an impressive array of methods, including in vivo measurements, EMG, blood samples, IHC and RNAseq.

We thank the Reviewer for the positive feedback regarding our work, it is truly appreciated.

The following findings have, to this reviewer's knowledge, previously been reported in relation to disuse:

- TTP63%, reflective of rapid force production, worsen with disuse (Monti et al., 2021).
- Activation capacity decline with disuse (Suetta et al., 2009).
- MUP properties change with disuse (Campbell et al., 2019).
- Muscle morphology change following disuse and retraining (Hvid et al., 2010)
- Number of NCAM+ fibers increase following disuse (Arentson-Lantz et al., 2016; Demangel et al., 2017; Monti et al., 2021)
- CAF increase following disuse (Monti et al., 2021)

The following findings are either completely novel (to this reviewer's knowledge) or oppose previous findings:

- Neurofilament light chain is increased with disuse and might serve as a useful biomarker for nerve damage.
- NMJ dysfunction has been reported following disuse (Grana et al., 1996), but in the present study the authors seen no changes in NMJ stability. Grana et al., also used a 4-week cast immobilization model.
- RNAseq was also used in the groups previous paper, Monti et al., 2021. However, in that paper only a handful of targets were reported. The data in the present study is therefore more comprehensive.
- The percentage of enclosed type II fibers was reduced with retraining, although this was, as pointed out by the authors, likely tied with the change in type II percentage.

We thank the Reviewer for the excellent overview of the previous findings in the literature. We would like to specify that, in addition to the observation pointed out by the Reviewer, the search for regenerating neonatal myosin-positive fibres, fibre variability and the focus on skeletal muscle ion channels represent novelties of our work. Moreover, we would like to highlight that our study presents several novel findings also regarding motor unit properties: indeed, previous studies (Duchateau & Hainaut, 1990; Seki *et al.*, 2001, 2007) reported an altered firing rate modulation (only in small hand muscles) but in absence of information regarding MUP and NF MUP complexity and size.

To add to this, retraining following disuse is not always investigated, which greatly increases the

value of this study. Furthermore, the study overall seems to be well conducted and great expertise within the different methods are found among the list of authors.

We thank the Reviewer again for the positive feedback, we are glad that she/he appreciated our work.

There are several points, organized in a section-by-section manner, that I would like the authors to address. I truly hope that the authors can see reason in these comments, and that the comments will increase the value of the work.

We did our best to amend the manuscript and we truly believe that the Reviewer's comments improved the quality of our work.

Introduction

It should be pointed out that the study by Nishimune et al., mostly refers to animal studies, especially when it comes to pre and post synaptic alterations by exercise: "Physical exercise is well known to maintain NMJ integrity both at its pre- and post-synaptic components, via the action of different neurotrophins (Nishimune et al., 2014)."

Amended, as suggested.

Methods

I suggest that the authors consider adding a timeline (LS0, LS10, AR10 and AR21) to increase the readability of the study.

We thank the reviewers for the suggestion, a timeline has been provided in the graphical abstract.

How were the participants recruited and from where? Also, the authors argue that only males were recruited due to increased risk of deep venous thrombosis in females, but surely young and healthy women would be able to complete this protocol safely? Without having checked their references (Bleeker et al., 2004), I would image that health status in general would be more important than sex, when dealing with young individuals.

We thank the Reviewer for allowing us to clarify this point. Participants were recruited mainly via social media and through oral presentations at the end of university lectures.

We believe that reasonably, a young and healthy woman would be able to complete the protocol safely, but it should be considered that the risk for deep venous thrombosis could be still higher for a woman compared to a young male individual (see the clear figure B below from (Roach *et al.*, 2014)), particularly if she is assuming oral contraceptive pills, known to increase blood coagulation (Bonnar, 1987).

Figure B: The absolute risk of first venous thrombosis in men and women across the lifespan (from (Roach *et al.*, 2014)).

Please provide the 1RM data, which I assume was measured 3-4 times in total during retraining. If the training data was logged this could also be plotted. This would give an idea of the relative starting point of the legs and how the "disused" leg would catch up.

In Figure C you can find the unilateral 1RM progression for both leg press (panel A) and leg extension (panel B) exercises. However, we believe that these data are supplementary in nature and we would prefer presenting them just in this open peer review history.

Figure C: Progression in unilateral one-repetition maximum (1RM) for leg press (A) and leg extension (B) exercise during the active recovery phase. 1RM was evaluated during the first session of each training week. * $p < 0.05$; ** $p < 0.01$; *** $p < 0.001$

Regarding ultrasound measurements, why did you decide to average the measurements at 30, 50 and 70% femur length, rather than showing these lengths individually, as you have previously done for the hamstrings (Sarto et al., 2021; Franchi et al., 2022)? Also, how many scans were performed at each site, as if more than one how was this dealt with?

In the present work we decided to report only the average of the measurements at 30, 50 and 70% femur length for two reasons: (i) for clarity - differently from our previous works on hamstrings that were almost entirely focused on morphological adaptations (Sarto *et al.*, 2021; Franchi *et al.*, 2022), in the present study we report the outcomes of a broad range of methods; thus we thought that the readers could be confused by multiple measures of muscle size reported for both quadriceps and vastus lateralis; (ii) for having a single morphological measure that accounts for regional differences in muscle size.

Two scans were obtained for each site and the image with the best quality was analysed.

What software was used for DQEMG analysis?

The name of the software employed is actually "DQEMG", developed and described by Prof. Daniel Stashuk (Stashuk, 1999).

The biopsy method proposed by Aubertin-Leheudre, where the intent is to increase the number of NMJs obtained per biopsy, was used in the present manus. Was this done, because the authors at some point intended to analyze NMJs from humans using this method, but perhaps did not get enough biopsies with NMJs? The rationale for choosing this method should at least be clear. Also, in the original publication, Aubertin-Leheudre et al., report to get ~40 NMJs per biopsy as opposed to ~0 when using normal method (Aubertin-Leheudre et al., 2020). The authors in the present study argue, in relation to the RNAseq and IHC data, that these genes and proteins that they measure are upregulated extra-synaptically when the muscle fiber is denervation/destabilized. This I would normally agree with, but since the special biopsy technique was used, this claim is problematic, as some of the "positive" signal could in fact come from actual NMJs in the biopsies? Please consider this in your discussion AND quantify NMJs in your biopsy sections using α -bungarotoxin and/or nestin (no graph needed, just the number of NMJs per biopsy) (Soendenbroe et al., 2022).

We thank the Reviewer for allowing us to discuss this; we believe this is really an important point. A biopsy sampling method similar to the one by Aubertin-Leheudre et al. (Aubertin-Leheudre *et al.*, 2019) was employed for two different reasons: (i) since our intention was to relate the changes in iEMG measures to biopsy outcomes, we thought that collecting both in the same region (close to the motor point) was the best choice; (ii) as the Reviewer guessed, we originally intended to analyse also NMJs morphology for this work. An additional biopsy part (~30/40mg) was entirely dedicated to this aim, fixed in paraformaldehyde and stained with α -bungarotoxin. Unfortunately, we did not find NMJs in a sufficient number of biopsies: 4/11 at LS0; 1/11 at LS10 (not in one of the participants in which it was found also at baseline) and 4/9 at AR21; for a total of 9 out of 31 biopsies (29%). Our success rate was lower compared to the one reported by Aubertin-Leheudre et al (53%) but it should be considered that we dedicated to this analysis a part of biopsy that was much smaller (~30/40mg vs ~150 mg). Analysis of the NMJ morphology was then not performed.

Regarding Reviewer's concern about the impact of this sampling method on our results, we believe it is unlikely that this has a significant impact on our results. Firstly, the biopsy part that was used for RNAseq analysis was very small (~10/15mg) and this makes the detection of NMJs even more

difficult. Thus, even if some NMJs were present in some samples, we believe that the large majority of myonuclei that contributed to RNAseq analysis can be considered extra-synaptic and the total pool of myonuclei was heterogeneous and comprehensive. Secondly, as explained above, we found a higher number of biopsies with NMJs at LS0 compared to LS10. Thus, we believe it is unlikely that some of our results following ULLS (i.e. upregulation of ACh receptors genes, increased percentage of NCAM positive fibers etc.) are the "positive" signals of the actual presence of NMJs in the biopsies.

Finally, there is some ambiguity when it comes to biopsy sampling. Biopsies were sampled at three time points, but it is not clear whether the order was randomized or fixed, and how close were the sample sites were to one another? The authors also mention that VL usually have three distinct motor points, but they are spatially separated, with some of them located at a place where biopsies would not be taken from (Gobbo et al., 2014). So, assuming that biopsies at all time points were taken ~ 2 cm from the same motor point, the distance between sites might be too little avoid effects of pre-sampling (Vissing et al., 2005). And perhaps this could also influence on iEMG measurements, since they too were performed close to the motor point. All these elements have to be considered and specified in the text, and some sort of overview figure showing exactly how these things were performed on each thigh, would be greatly appreciated.

All three biopsies were performed at ~ 2 cm from the same central motor point, with a distance of $\sim 2/3$ cm between them in order to avoid the effects of pre-sampling. Order of Biopsy 2 and 3 was randomized. A graphical illustration is provided in Figure D.

Figure D: Graphical illustration of the biopsy sampling locations.

Add RRID for NCL-MHCd. AB_563901 I think?

Amended, as suggested.

I appreciate the effort to run positive controls, but I do not understand, from your description, how an antibody made in mouse (NCL-MHCd) was used on mouse tissue? Please specify.

We thank the Reviewer for noticing this, we used regenerating rat tissue, not mouse. We amended that in the manuscript.

What objective (and numerical aperture) was used?

The objective used was 20x with 0.4 of numerical aperture. We added this information to the manuscript.

How was imaging performed in terms of exposure times etc., fixed or variable conditions?

We did not use fixed conditions since it was not our aim to perform a comparison on the signal intensity. Thus, we employed variable conditions.

These sentences say the same thing: "Negative controls were performed by omitting the primary antibodies from sample incubations." AND "...while negative controls were performed by omitting the primary antibodies from sample incubations"

We thank the Reviewer for spotting this. We deleted the first sentence, as suggested.

Were biopsy analyses (IHC, morphology) performed blinded for leg/time point?

Yes, the analysis was blinded. We added this information to the manuscript.

The following description of enclosed fibers is not entirely clear to me, considering the results are shown as % grouped fibers out of all fibers: "For the quantification of large and very large fibre type clusters (>10 or >20 fibres, respectively), the method of "enclosed fibre"". Also, please be consistent in the use of "grouped" versus "enclosed", as the former seems to refer to fibers of the same type that are connected to each other, while the latter refer to a fiber that is complete surround by fibers of its own type.

"Enclosed fibres" is the criteria (Jennekens *et al.*, 1971) used to define a type grouping since not all the grouped fibres represent a type grouping. We amended the text in the manuscript to make this point clearer.

Also, fiber diameter variability was also assessed. However, another measurement, usually annotated "shape factor", has previously been used to analyze variability in muscle morphology (Barnouin *et al.*, 2017; Messa *et al.*, 2020). Did the authors consider using this measurement?

We thank the Reviewer for the suggestion. The shape factor is an index of the roundness of the fiber, calculated as follows: $\text{perimeter}^2 / (4\pi \times \text{FCSA})$. An increased shape factor represents increased angularity (Barnouin *et al.*, 2017; Messa *et al.*, 2020). This parameter seems very interesting for an evaluation of changes in myofibre shape, however, we think that this does not represent a measurement of fiber diameter variability. Indeed, in both the cited papers (Barnouin *et al.*, 2017;

Messa *et al.*, 2020), fiber size variability is simply expressed as the standard deviation of the fibre CSA.

In relation to the RNAseq, I appreciate that it has been performed by a third-party company, which is okay. But there are central parts of the method section, which is simply too unspecific, for example in relations to products used (cat. number, amount, etc.). Please specify as much as possible.

We added as much information as possible to the text.

Importantly, as this reviewer understands it, it is only genes that are "known" to be involved in NMJ and skeletal muscle ion channel regulation, that is shown in the data sets? In other words, a fraction of all the genes that were detected was chosen for this analysis. While it is not in my place to decide whether this approach is correct, I believe that this should be stated in VERY clear terms in both the methods, results and discussion sections and in corresponding figure legends. And it would be of value to say something about how large/small this fraction was?

As we explained to the Reviewer 1, since the scope of this work is hypothesis-driven (as opposed to an exploratory, fully data-driven approach) we did not expand the pathway characterization to all other pathways available (e.g from the Molecular Signatures database, including Gene Ontology).

We added the number of genes involved in each gene set. We also uploaded the details of NMJ (Data S1) and ion channels (Data S2) RNAseq results to the Supporting Information section.

Please elaborate and specify the GSEA approach.

We added details on the GSEA approach implemented in our analysis.

Please provide references for neurofilament light chain to be a "well-established" marker of axonal damage. Again, the level of detail on this method is very low (company analysis), so please specify as much as possible.

We provided two references in support of neurofilament light chain as a biomarker of axonal damage both in aging and neurological disorders scenarios.

We asked the company for further details regarding SIMOA analysis and we added them to the manuscript. Now it reads:

“SIMOA analysis was also performed to evaluate neurofilament light chain concentration, a well-established biomarker of axonal damage, previously employed both in aging and neurological disorders scenarios (Khalil *et al.*, 2018; Pratt *et al.*, 2022). The samples were submitted to the SIMOA service offered by Wieslab AB, a Svar Life Sci company, Lundavägen MALMO (Sweden); in accordance with Good Laboratory Practice (GLP) principles. SIMOA analysis was performed on a Simoa HD-X Analyzer (PN 10041537) supplied by Quanterix in agreement with standard protocol suggested by Quanterix. In particular, serum samples were diluted 1:4 and measurements obtained in double replicates. The kit used for the reported analysis was Simoa NF-light Advantage Kit HD-1/HD-X (Item 103186; Lot 502845).”

Why was CAF used for power calculations, and not a more classic measurement, like MVC or mass?

As we explained, in the present study we were particularly interested in studying NMJ alterations with unloading, thus, it was felt more appropriate to focus the power calculation on CAF compared

to more classic functional or morphological measurements.

In relation to the statistical analyses of the EMG data, it seems that the authors have used a common approach with this type of data, which is use an average of ALL motor unit measurements pooled together, and not per subject, which greatly increases the n. They use a generalized linear mixed effect model (fixed effect: time; cluster variable: subject) and provide a reference in an attempt to justify this (Yu et al., 2022). However, it does not seem appropriate to treat a single MU as a statistical unit, instead it should be a single individual. If this was done in a biopsy analysis, n's of several thousands would not be uncommon when evaluated changes in for instance fiber size. I suggest that the authors 1) consider these elements and their limitations in their discussion of the data, 2) specify in the figure legends exactly how statistical testing was performed, and 3) provide complimentary figures and statistical outcomes of these data sets (EMG measurements) performed with subject as the statistical unit. This last element can figure in the response letter only, if desired.

We thank the Reviewer for the observation. We would like to specify that our generalized linear mixed models do not consider an average of all motor unit measurements. Indeed, being “subject” the cluster variable, our statistical unit is the single individual and not the single motor unit. Our model is suitable for data of this nature as it: i) considers the whole population of sampled decomposed motor units and not just the mean values obtained from each participant, which preserves better the variability within and across individuals and ii) handles missing data better than traditional ANOVA framework.

We believe that this does not represent a limitation of our work, as this statistical approach is considered more rigorous compared to traditional statistics such as ANOVAs (Brown, 2021; Yu *et al.*, 2022). Moreover, the large majority of the new works in the field of motor unit physiology employ and support a generalized linear mixed models approach (for instance: Guo *et al.*, 2021, 2022; Hassan *et al.*, 2021; Jones *et al.*, 2021; Orssatto *et al.*, 2021; Mesquita *et al.*, 2022)

We appreciate the suggestion of the Reviewer to specify the statistical approach used in the figure legends and we amended the manuscript accordingly.

Below figures (E-H) of iEMG results analysed with mixed-effect ANOVAs are provided but, again, we believe that our approach is the most correct.

Figure E: Changes in MUP properties obtained at 10% of maximum voluntary contraction using intramuscular electromyography after 10 days of unilateral lower limb suspension (LS10) and 21 days of active recovery (AR21). Statistical analysis was performed using one-way repeated measures mixed-effects ANOVA. Results are shown as mean and standard deviation. Motor unit potential (MUP) area (A); MUP duration (C); MUP turns (C) and mean interdischarge interval (IDImean) (D). LS0: baseline data collection; * $p < 0.05$; *** $p < 0.001$

Figure F: Changes in near fibre (NF) electromyography outcomes at 10% of maximum voluntary contraction after 10 days of unilateral lower limb suspension (LS10) and 21 days of active recovery (AR21). Statistical analysis was performed using one-way repeated measures mixed-effects ANOVA. Results are shown as mean and standard deviation. Near fibre motor unit potential (NFM) area (A); NFM duration (B); NF count (C); NFM jiggle (D) and NFM segment jitter (E). LS0: baseline data collection; ** $p < 0.01$

Figure G: Changes in MUP properties obtained at 25% of maximum voluntary contraction using intramuscular electromyography after 10 days of unilateral lower limb suspension (LS10) and 21 days of active recovery (AR21). Statistical analysis was performed using one-way repeated measures mixed-effects ANOVA. Results are shown as mean and standard deviation. Motor unit potential (MUP) area (A); MUP duration (C); MUP turns (C) and mean interdischarge interval (IDI_{mean}) (D). LS0: baseline data collection; * $p < 0.05$

Figure H: Changes in near fibre (NF) electromyography outcomes at 25% of maximum voluntary contraction after 10 days of unilateral lower limb suspension (LS10) and 21 days of active recovery (AR21). Statistical analysis was performed using one-way repeated measures mixed-effects ANOVA. Results are shown as mean and standard deviation. Near fibre motor unit potential (NFM) area (A); NFM duration (B); NF count (C); NFM jiggle (D) and NFM segment jitter (E). LS0: baseline data collection; *p<0.05

Results / figures

Please write statistical test in figure legend, especially since regular or mixed effects were used depending on missing values.

Amended, as suggested.

"Interestingly, for both muscles, CSA mean at AR21 was higher than LS0 and AR21...". Both muscles here refers to vastus lateralis and quadriceps, but the latter is not a muscle. Also, it seems it should be LS10 and not AR21 in that sentence?

We thank the Reviewer for noticing this, we amended the manuscript accordingly.

Figure 2+3: Show individual subject values. Also, could the authors provide some representative

curves for both MUP, NF and NMJ stability. Please also specify within curves what each measurement refers to. See these papers for examples (Piasecki et al., 2021; Guo et al., 2021).

We thank the Reviewer for the suggestion. We intentionally did not show individual subject values in Figure 2 and 3 (iEMG outcomes), as in our previous recent J Physiol work (Monti *et al.*, 2021). Indeed, generalized linear mixed models are not based on the mean of the subjects' values; but on estimated marginal means that are not individual (one estimate for time point).

We added representative MUP and NF waveforms for each time point in Figure 2 and Figure 3 of the manuscript.

Muscle morphology. Please also provide fiber CSA instead of only ferret diameter. Representative images of ATPase staining would be greatly appreciated.

We believe that Ferret diameter approach is superior to fiber CSA in human biopsies for two reasons: (i) the coefficient of variation is lower (Briguet *et al.*, 2004) and (ii) it is less sensitive to the distortion of obliquely cut or kinked muscle fibers (Dubowitz, 1985).

We added a representative ATPase image to Figure 7, as suggested.

Please show the NCAM and MyHCn data. Use individual values and median, given the lack of normal distribution. Please also provide representative images.

We thank the Reviewer for the suggestion.

NCAM:

As explained in the Methods section, NCAM evaluation was part of a parallel investigation focused only on the unloading period (paper in submission). For this reason, we would prefer to not add the representative images and individual values to the main text, but we gladly report them here (see Table A and Figure I).

ID	LS0	LS10
S1	neg	neg
S2	0,2%	3,5%
S3	neg	neg
S4	neg	2,3%
S5	neg	4,5%
S6	neg	0,7%
S7	neg	neg
S8	0,3%	1,3%
S9	neg	neg
S10	neg	0,4%
S11	neg	neg
MEAN	0.045	1.155
MEDIAN	0.000	0.400
SD	0.104	1.596

Table A: Individual raw NCAM values at the different time points. LS0: baseline; LS10: after 10 days of unilateral lower limb suspension; SD: standard deviation

Figure I: Representative NCAM image pre- and post-unilateral lower limb suspension.

Neonatal myosin:

Neonatal myosin-positive fibres were not detected at LS0, LS10 and AR21, therefore the median is 0 at any time point and we cannot provide a representative image from the present study. However, in Figure L the Reviewer can find representative MyHCn images from Dr. Zampieri's Laboratory.

Fig L: Representative image of a tissue section stained for laminin and nMyHC (A). Positive control for MyHC staining on a tissue section obtained from a regenerating muscle of a rat (B). No nMyHC fibres were detected in human muscle biopsies at LS0, LS10 and AR21. Colour legend: Laminin (red); nMyHC (green); DAPI (blue)

Figure 4+5+6. Please name all significant genes or provide a corresponding list of significant genes. Right now the majority of information is lost as many genes remain unspecified.

We uploaded the details of NMJ (Data S1) and ion channels (Data S2) RNA-Seq results to the Supporting Information section. The whole RNA-Seq dataset will be available at the Gene Expression Omnibus repository upon publication of this article.

Figure 5: It is not clear among the clustered targets in the middle which gene name belongs to which dot. Consider using a small line between name and dot. Also, please check whether text and figure legend are aligned in terms of what is down- and upregulation in the four fields.

We thank the Reviewer for the suggestion, we edited the figure adding some lines between name and dot. Also, we amended the text and figure legends.

Figure 7: The significant increase in CAF (5.4 %, $p=0.038$) seems very slim. Please provide the individual data points used in the statistical analysis in the response. Also, E and F should have the same y-axis.

We agree that the CAF increase is modest and less pronounced compared to our previous work on bed rest (5.4% vs 19.2%, respectively) (Monti *et al.*, 2021). This is most likely due to the lower muscle mass undergoing unloading (one leg in ULLS versus whole body in bed rest), as explained in the manuscript.

Individual raw CAF values are provided in Table B.

ID	LS0	LS10	AR21
S1	3782.72	3536.83	3783.82
S2	5057.10	5332.55	4117.13
S3	4343.70	4707.13	4476.62
S4	4145.69	4627.48	4902.75
S5	4360.08	4984.23	4874.38
S6	4715.55	4905.41	4481.84
S7	3158.57	3203.52	3781.89
S8	4624.28	4765.35	5739.28
S9	2690.42	2797.62	2604.19
S10	4982.58	5662.20	6139.87
S11	4900.93	4916.75	5352.72
MEAN	4251.06	4494.46	4568.59
SD	828.66	932.36	998.65

Table B. Individual raw CAF values at the different time points. LS0: baseline; LS10: following 10 days of unilateral lower limb suspension; AR21: following 3 weeks of active recovery; SD: standard deviation

We amended the y axis of the Figure 7, panel F.

Discussion

Please discuss why only young were studied, as an old group in my view would greatly enhance the relevancy of the work?

We completely agree with the Reviewer, including a group of old individuals would have been a great addition. However, please consider that the study was conducted after the second COVID

pandemic wave. Compared to young individuals, older adults were more concerned about their health status and thus less prone to visit the laboratory for so many appointments (one for familiarization, three for data collection and nine for training sessions).

As the authors note themselves, there is a very dramatic reduction in type II fiber percentage (corresponding increase in type I). Since ATPase was performed to only distinguish between type I and II, it is not clear whether this results from isoform specific changes. Is the dramatic change related to altered coexpression of isoforms or perhaps a reduction in type IIx? (Andersen et al., 1999 p.19; Andersen & Aagaard, 2000).

We believe that the reduction in type II fibres percentage with active recovery may be mainly due to changes in the coexpression relationship between type I and type II fibres. Single fiber proteomics studies (Murgia *et al.*, 2017, 2021) showed that “pure fibers”, (i.e. containing at least 80% of either MYH7, marker of slow type 1 fibers, MYH2, marker of fast 2A fibers, or MYH1, marker of fast 2X fibers) are rare, thus little changes induced by resistance training may be sufficient to induce the shift toward the predominantly fast group to the predominantly slow group in some muscle fibres.

The percentage of enclosed type II fibers seems very high at baseline (LS0), considering this is young healthy men. Messa et al., 2020, reported, using the same method by Jennekens et al., 1971, a percentage of type II enclosed fibers of ~2% in young male (n=14) and female (n=8) controls (Jennekens et al., 1971; Messa et al., 2020). Please discuss.

We thank the Reviewer for raising this point of discussion. We believe that our percentage of enclosed type II fibres is higher compared to the study by Messa et al (Messa *et al.*, 2020) probably due to the high percentage of fast fibre (~66% at LS0; ~68% at LS10) observed in the present work, considering that the vastus lateralis is generally considered a mixed muscle (~50% fast fibre). Our findings could be due to individual variability and/or biopsy sampling location (close to the motor point) and characteristics (e.g. slightly different depths) (Lexell *et al.*, 1983; Horwath *et al.*, 2021).

In relation to NMJ stability, the authors base this only on NF jitter and jiggle, but would the other NF EMG measurements not also inform on this? Please explain.

The essential difference between NF MUP segment jitter and jiggle and the other NF EMG measurements is that the latter are based on characteristics of a template or representative waveform, while NF MUP segment jitter and jiggle are computed across all of the NF MUPs of an extracted MUPT. Thus, NF MUP segment jitter and jiggle reflect NF MUP temporal and shape variability/instability, respectively, and are thought to reflect NMJ transmission stability (i.e. NMJ function). Other NF EMG parameters provide a different kind of information: for instance, NFM area and NF count reflect MU muscle fibre density (i.e. size and number of MU fibres in a particular area, close to the tip of the recording electrode).

"Similarly, muscle size, NMJ stability, MUP and NF MUP characteristics were restored within 21 days of AR.". It does not make sense that it has just been concluded that NMJ stability was not affected by disuse, yet now it is restored?

We thank the Reviewer for pointing this out. We were referring to the NMJ molecular stability (changes in NMJ-related genes). We amended the manuscript accordingly.

"Overall, the AR period counteracted the effects of unloading on the NMJ and ion channel gene expression.". Since there was no non-exercising control group, it can in fact no be concluded that it

was AR that counteracted the effects of unloading, it could also be time itself. This point should perhaps be brought forward by the authors, and perhaps they know of other studies that have compared active and passive recovery following disuse. Suetta et al., 2004 compared standard recovery with strength training + standard recovery in elderly men and women following hip-replacement surgery and found that exercise was crucial in order to recover strength (Suetta et al., 2004). But then again, this was elderly people, following surgery, presumably in pain and therefore perhaps not relevant to this study.

We thank the Reviewer for the observation; we decided to reword the sentence. We are not aware of other studies that compared active and passive recovery following disuse but we agree with the Reviewer that this is a very important point.

References

- Aubertin-Leheudre M, Pion CH, Vallée J, Marchand S, Morais JA, Bélanger M & Robitaille R (2019). Improved Human Muscle Biopsy Method To Study Neuromuscular Junction Structure and Functions with Aging. *Journals Gerontol Ser A* **75**, 2098–2102.
- Barnouin Y, McPhee JS, Butler-Browne G, Bosutti A, De Vito G, Jones DA, Narici M, Behin A, Hogrel JY & Degens H (2017). Coupling between skeletal muscle fiber size and capillarization is maintained during healthy aging. *J Cachexia Sarcopenia Muscle* **8**, 647–659.
- Bonnar J (1987). Coagulation effects of oral contraception. *Am J Obstet Gynecol* **157**, 1042–1048.
- Briguet A, Courdier-Fruh I, Foster M, Meier T & Magyar JP (2004). Histological parameters for the quantitative assessment of muscular dystrophy in the mdx-mouse. *Neuromuscul Disord* **14**, 675–682.
- Brocca L, Cannavino J, Coletto L, Biolo G, Sandri M, Bottinelli R & Pellegrino MA (2012). The time course of the adaptations of human muscle proteome to bed rest and the underlying mechanisms. *J Physiol* **590**, 5211–5230.
- Brocca L, Longa E, Cannavino J, Seynnes O, de Vito G, McPhee J, Narici M, Pellegrino MA & Bottinelli R (2015). Human skeletal muscle fibre contractile properties and proteomic profile: Adaptations to 3 weeks of unilateral lower limb suspension and active recovery. *J Physiol* **593**, 5361–5385.
- Brown VA (2021). An introduction to linear mixed-effects modeling in R. *Adv Methods Pract Psychol Sci*.
- Demangel R, Treffel L, Py G, Brioche T, Pagano AF, Bareille MP, Beck A, Pessemesse L, Candau R, Gharib C, Chopard A & Millet C (2017). Early structural and functional signature of 3-day human skeletal muscle disuse using the dry immersion model. *J Physiol* **595**, 4301–4315.
- Deschenes MR (2019). Adaptations of the neuromuscular junction to exercise training. *Curr Opin Physiol* **10**, 10–16.
- Dubowitz, V. Muscle Biopsies. In *A Practical Approach*, 2nd ed.; Saunders, W.B., Ed.; Baillière Tindall Publishers: London, UK, 1985; pp. 86–90).
- Duchateau J & Hainaut K (1990). Effects of immobilization on contractile properties, recruitment and firing rates of human motor units. *J Physiol* **422**, 55–65.
- Fitts RH, Trappe SW, Costill DL, Gallagher PM, Creer AC, Colloton PA, Peters JR, Romatowski JG, Bain

- JL & Riley DA (2010). Prolonged space flight-induced alterations in the structure and function of human skeletal muscle fibres. *J Physiol* **588**, 3567–3592.
- Franchi M V., Sarto F, Simunič B, Pišot R & Narici M V. (2022). Early Changes of Hamstrings Morphology and Contractile Properties During 10 Days of Complete Inactivity. *Med Sci Sport Exerc Publish Ah*, 1346–1354.
- Guo Y, Jones EJ, Inns TB, Ely IA, Stashuk DW, Wilkinson DJ, Smith K, Piasecki J, Phillips BE, Atherton PJ & Piasecki M (2022). Neuromuscular recruitment strategies of the vastus lateralis according to sex. *Acta Physiol* 1–14.
- Guo Y, Piasecki J, Swiecicka A, Ireland A, Phillips BE, Atherton PJ, Stashuk D, Rutter MK, McPhee JS & Piasecki M (2021). Circulating testosterone and dehydroepiandrosterone are associated with individual motor unit features in untrained and highly active older men. *GeroScience* 1215–1228.
- Hassan AS, Fajardo ME, Cummings M, McPherson LM, Negro F, Dewald JPA, Heckman CJ & Pearcey GEP (2021). Estimates of persistent inward currents are reduced in upper limb motor units of older adults. *J Physiol* **599**, 4865–4882.
- Horwath O, Envall H, Roja J, Emanuelsson EB, Sanz G, Ekblom B, Apro W & Moberg M (2021). Variability in vastus lateralis fiber type distribution, fiber size, and myonuclear content along and between the legs. *J Appl Physiol* **131**, 158–173.
- Hvid LG, Ørtenblad N, Aagaard P, Kjaer M & Suetta C (2011). Effects of ageing on single muscle fibre contractile function following short-term immobilisation. *J Physiol* **589**, 4745–4757.
- Jennekens FGI, Tomlinson BE & Walton JN (1971). Histochemical aspects of five limb muscles in old age an autopsy study. *J Neurol Sci* **14**, 259–276.
- Jones EJ, Piasecki J, Ireland A, Stashuk DW, Atherton PJ, Bethan E, McPhee JS & Piasecki M (2021). Lifelong exercise is associated with more homogeneous motor unit potential features across deep and superficial areas of vastus lateralis. *Geroscience* **43**, 1555–1565.
- Khalil M, Teunissen CE, Otto M, Piehl F, Sormani MP, Gattringer T, Barro C, Kappos L, Comabella M, Fazekas F, Petzold A, Blennow K, Zetterberg H & Kuhle J (2018). Neurofilaments as biomarkers in neurological disorders. *Nat Rev Neurol* **14**, 577–589.
- Korotkevich G, Sukhov V, Budin N, Shpak B, Artyomov MN & Sergushichev A (2021). Fast gene set enrichment analysis. *bioRxiv* 060012.
- Lexell J, Henriksson-Larsen K & Sjöström M (1983). Distribution of different fibre types in human skeletal muscles 2. A study of cross-sections of whole m. vastus lateralis. *Acta Physiol Scand* **117**, 115–122.
- Mesquita RNO, Taylor JL, Trajano GS, Škarabot J, Holobar A, Gonçalves BAM & Blazevich AJ (2022). Effects of reciprocal inhibition and whole-body relaxation on persistent inward currents estimated by two different methods. *J Physiol* **600**, 2765–2787.
- Messa GAM, Piasecki M, Rittweger J, McPhee JS, Koltai E, Radak Z, Simunic B, Heinonen A, Suominen H, Korhonen MT & Degens H (2020). Absence of an aging-related increase in fiber type grouping in athletes and non-athletes. *Scand J Med Sci Sport* **30**, 2057–2069.
- Monti E, Reggiani C, Franchi M V, Toniolo L, Sandri M, Armani A, Zampieri S, Giacomello E, Sarto F, Sirago G, Murgia M, Nogara L, Marcucci L, Ciciliot S, Šimunic B, Pišot R & Narici M V (2021). Neuromuscular junction instability and altered intracellular calcium handling as early determinants of force loss during unloading in humans. *J Physiol* **599**, 3037–3061.

- Murgia M, Nogara L, Baraldo M, Reggiani C, Mann M & Schiaffino S (2021). Protein profile of fiber types in human skeletal muscle: a single-fiber proteomics study. *Skelet Muscle* **11**, 1–19.
- Murgia M, Toniolo L, Nagaraj N, Ciciliot S, Vindigni V, Schiaffino S, Reggiani C & Mann M (2017). Single Muscle Fiber Proteomics Reveals Fiber-Type-Specific Features of Human Muscle Aging. *Cell Rep* **19**, 2396–2409.
- Orssatto LBR, Borg DN, Blazeovich AJ, Sakugawa RL, Shield AJ & Trajano GS (2021). Intrinsic motoneuron excitability is reduced in soleus and tibialis anterior of older adults. *GeroScience* **43**, 2719–2735.
- Piasecki M, Ireland A, Coulson J, Stashuk DW, Hamilton-Wright A, Swiecicka A, Rutter MK, McPhee JS & Jones DA (2016a). Motor unit number estimates and neuromuscular transmission in the tibialis anterior of master athletes: evidence that athletic older people are not spared from age-related motor unit remodeling. *Physiol Rep*.
- Piasecki M, Ireland A, Stashuk D, Hamilton-Wright A, Jones DA & McPhee JS (2016b). Age-related neuromuscular changes affecting human vastus lateralis. *J Physiol* **594**, 4525–4536.
- Pratt J, De Vito G, Segurado R, Pessanha L, Dolan J, Narici M & Boreham C (2022). Plasma neurofilament light levels associate with muscle mass and strength in middle-aged and older adults: findings from GenoFit. *J Cachexia Sarcopenia Muscle* **13**, 1811–1820.
- Roach REJ, Cannegieter SC & Lijfering WM (2014). Differential risks in men and women for first and recurrent venous thrombosis: The role of genes and environment. *J Thromb Haemost* **12**, 1593–1600.
- Sarto F, Monti E, Simunič B, Pišot R, Narici M V. & Franchi M V. (2021). Changes in Biceps Femoris Long Head Fascicle Length after 10-day Bed Rest Assessed with Different Ultrasound Methods. *Med Sci Sport Exerc* 1–9.
- Seki K, Kizuka T & Yamada H (2007). Reduction in maximal firing rate of motoneurons after 1-week immobilization of finger muscle in human subjects. *J Electromyogr Kinesiol* **17**, 113–120.
- Seki K, Taniguchi Y & Narusawa M (2001). Effects of joint immobilization on firing rate modulation of human motor units. *J Physiol* **530**, 507–519.
- Stashuk DW (1999). Decomposition and quantitative analysis of clinical electromyographic signals. *Med Eng Phys* **21**, 389–404.
- Widrick JJ, Romatowski JG, Norenberg KM, Knuth ST, Bain JLW, Riley DA, Trappe SW, Trappe TA, Costill DL & Fitts RH (2001). Functional properties of slow and fast gastrocnemius muscle fibers after a 17-day spaceflight. *J Appl Physiol* **90**, 2203–2211.
- Widrick JJ, Trappe SW, Romatowski JG, Riley DA, Costill DL & Fitts RH (2002). Unilateral lower limb suspension does not mimic bed rest or spaceflight effects on human muscle fiber function. *J Appl Physiol* **93**, 354–360.
- Yu Z, Guindani M, Grieco SF, Chen L, Holmes TC & Xu X (2022). Beyond t test and ANOVA: applications of mixed-effects models for more rigorous statistical analysis in neuroscience research. *Neuron* **110**, 21–35.

Dear Mr Sarto,

Re: JP-RP-2022-283381R1 "Effects of short-term unloading and active recovery on human motor unit properties, neuromuscular junction transmission and transcriptomic profile" by Fabio Sarto, Dan Stashuk, Martino V Franchi, Elena Monti, Sandra Zampieri, Giacomo Valli, Giuseppe Sirago, Julian Candia, Lisa Hartnell, Matteo Paganini, Jamie S McPhee, Giuseppe De Vito, Luigi Ferrucci, Carlo Reggiani, and Marco Narici

Thank you for submitting your manuscript to The Journal of Physiology. It has been assessed by a Reviewing Editor and by 2 expert Referees and I am pleased to tell you that it is considered to be acceptable for publication following satisfactory revision.

The reports are copied at the end of this email. Please address all of the points and incorporate all requested revisions, or explain in your Response to Referees why a change has not been made.

NEW POLICY: In order to improve the transparency of its peer review process The Journal of Physiology publishes online as supporting information the peer review history of all articles accepted for publication. Readers will have access to decision letters, including all Editors' comments and referee reports, for each version of the manuscript and any author responses to peer review comments. Referees can decide whether or not they wish to be named on the peer review history document.

Authors are asked to use The Journal's premium BioRender (<https://biorender.com/>) account to create/redraw their Abstract Figures. Information on how to access The Journal's premium BioRender account is here: <https://physoc.onlinelibrary.wiley.com/journal/14697793/biorender-access> and authors are expected to use this service. This will enable Authors to download high-resolution versions of their figures. The link provided should only be used for the purposes of this submission. Authors will be charged for figures created on this premium BioRender account if they are not related to this manuscript submission.

I hope you will find the comments helpful and have no difficulty returning your revisions within 4 weeks.

Your revised manuscript should be submitted online using the links in Author Tasks Link Not Available.

Any image files uploaded with the previous version are retained on the system. Please ensure you replace or remove all files that have been revised.

REVISION CHECKLIST:

- Article file, including any tables and figure legends, must be in an editable format (eg Word)
- Abstract figure file (see above)
- Statistical Summary Document
- Upload each figure as a separate high quality file
- Upload a full Response to Referees, including a response to any Senior and Reviewing Editor Comments;
- Upload a copy of the manuscript with the changes highlighted.

- A potential 'Cover Art' file for consideration as the Issue's cover image;
- Appropriate Supporting Information (Video, audio or data set https://jp.msubmit.net/cgi-bin/main.plex?form_type=display_requirements#supp).

To create your 'Response to Referees' copy all the reports, including any comments from the Senior and Reviewing Editors, into a Word, or similar, file and respond to each point in colour or CAPITALS and upload this when you submit your revision.

I look forward to receiving your revised submission.

If you have any queries please reply to this email and staff will be happy to assist.

Yours sincerely,

Scott K. Powers
Senior Editor
The Journal of Physiology
<https://jp.msubmit.net>
<http://jp.physoc.org>
The Physiological Society
Hodgkin Huxley House
30 Farringdon Lane
London, EC1R 3AW
UK
<http://www.physoc.org>
<http://journals.physoc.org>

REQUIRED ITEMS:

-You must start the Methods section with a paragraph headed Ethical Approval. If experiments were conducted on humans confirmation that informed consent was obtained, preferably in writing, that the studies conformed to the standards set by the latest revision of the Declaration of Helsinki, and that the procedures were approved by a properly constituted ethics committee, which should be named, must be included in the article file. If the research study was registered (clause 35 of the Declaration of Helsinki) the registration database should be indicated, otherwise the lack of registration should be noted as an exception (e.g. The study conformed to the standards set by the Declaration of Helsinki, except for registration in a database.). For further information see: <https://physoc.onlinelibrary.wiley.com/hub/human-experiments>

EDITOR COMMENTS

Reviewing Editor:

Both reviewers reiterated their positive comments on the manuscript. Both reviewers were mostly satisfied with the changes made on the original version of the manuscript. Both reviewers have a few relatively minor comments to make on the revised version, which the authors should take into consideration.

Senior Editor:

Thank you for revising your report. Your revised manuscript has (again) been reviewed by two referees and a review editor. The reviewers are pleased with your revisions but a few minor points remain unresolved. Please consider the reviewer suggestions carefully and revise your manuscript for re-review. We look forward to receiving your revised report.

REFEREE COMMENTS

Referee #1:

Dear authors, many parts of the paper have been improved. There remain a few key areas that may benefit from further consideration. I tried to highlight these points and are numbered below.

1) Major conclusions.

I note the authors response to first reviewer's overview.

Based on DQEMG measures, the authors stress in the abstract and key points (third bullet) of the manuscript that human NJM transmission stability is resilient. In the abstract and key points, this finding should be qualified that only the low-

threshold MU population was tested. There is no mention of low intensity voluntary contractions (up to 25% of MVC) or that lower-threshold MUs were sampled. This is now only mentioned in the discussion.

As the authors appreciate, there is a technical difference, because the blood, molecular and fibre type assays surveyed the MU population without a voluntary MU recruitment bias, whereas the DQEMG technique only recorded data from the low-threshold MU population.

In response to the acute disuse paradigm (LS10 as compared to LS0) muscle biopsies sampled the range between low- and high-threshold MU populations, and showed significant gene expression and fibre type distribution differences. Whereas DQEMG only sampled the low-threshold MU population and showed no difference in NMJ transmission stability.

Therefore, could this technical difference reasonably explain how the NMJ molecular alterations were observed in the absence of DQEMG NMJ transmission stability changes? And taken a step further, suggest different adaptive responses in low- vs. mixed-threshold MU populations?

2) Major conclusions/physiological mechanisms in humans.

I note the authors response to the first reviewer's comment number 21, subsection 1.

In human models of chronic disuse, such as ageing, there remains evidence that NMJs are stable across lifespan (Jones et al., 2017, Cell Reports 21:2348-2356). Likewise there is indirect support from EMG methods, like those used in the current manuscript, that the low-threshold, slow-type MU population has NMJ transmission resilience from chronic disuse in ageing (see reviews: Hepple and Rice, 2016, Journal of Physiology 594:1965-1978; Allen et al., 2021, Experimental Gerontology, 152:111465) and motor axon injury (example, Krarup et al., 2016, Clinical Neurophysiology 127:1675-1682). From this, we would expect that the low-threshold MU would be more likely to survive insult and re-innervate muscle fibres that become denervated.

With this view, without changes of MUP area or NF MUP jitter and jiggle at LS10 (acute disuse) does this provide support for NMJ functional (i.e., NMJ transmission stability) resilience? Can this NMJ functional resilience to acute disuse be generalized across both the low- and high-threshold MU populations as done in the abstract and key findings?

Referee #2:

The authors have replied and amended many of my suggested changes. At places where they have not made changes mostly comes down to personal preferences/beliefs (fiber diameter or CSA) and whether a specific type of statistics is suitable for the data (EMG data). I do not have enough insight to judge the latter, but this comment does not seem reasonable to me: "i) considers the whole population of sampled decomposed motor units and not just the mean values obtained from each participant"

Three points that are still lacking:

The biopsy sampling sites seem very closely located. This is important information to provide to the readers. 2-3 cm normally refers to longitudinal spacing, and from the image it does not seem to be 2-3 cm between sites. Sampling from prior biopsy sites is associated with altered gene expression.

The authors did not follow up on my recommendation to do a quick a-BTX staining (very simple, easily done), which could

have increased confidence in the RNAseq data not being confounded by NMJs.

Lastly, the authors refrain from showing the NCAM data in the manuscript, which is problematic, as they refer to them in the results and abstract: "and denervation status were assessed from blood samples and VL biopsies.". They should either remove completely or show data.

END OF COMMENTS

1st Confidential Review

29-Jul-2022

EDITOR COMMENTS

Reviewing Editor:

Both reviewers reiterated their positive comments on the manuscript. Both reviewers were mostly satisfied with the changes made on the original version of the manuscript. Both reviewers have a few relatively minor comments to make on the revised version, which the authors should take into consideration.

Senior Editor:

Thank you for revising your report. Your revised manuscript has (again) been reviewed by two referees and a review editor. The reviewers are pleased with your revisions but a few minor points remain unresolved. Please consider the reviewer suggestions carefully and revise your manuscript for re-review. We look forward to receiving your revised report.

We thank the Reviewing and Senior Editors for coordinating the review process. We did our best to amend the manuscript considering the last few minor points that remained unresolved.

REFEREE COMMENTS

Referee #1:

Dear authors, many parts of the paper have been improved. There remain a few key areas that may benefit from further consideration. I tried to highlight these points and are numbered below.

1) Major conclusions.

I note the authors response to first reviewer's overview.

Based on DQEMG measures, the authors stress in the abstract and key points (third bullet) of the manuscript that human NJM transmission stability is resilient. In the abstract and key points, this finding should be qualified that only the low-threshold MU population was tested. There is no mention of low intensity voluntary contractions (up to 25% of MVC) or that lower-threshold MUs were sampled. This is now only mentioned in the discussion.

As the authors appreciate, there is a technical difference, because the blood, molecular and fibre type assays surveyed the MU population without a voluntary MU recruitment bias, whereas the DQEMG technique only recorded data from the low-threshold MU population.

In response to the acute disuse paradigm (LS10 as compared to LS0) muscle biopsies sampled the range between low- and high-threshold MU populations, and showed significant gene expression and fibre type distribution differences. Whereas DQEMG only sampled the low-threshold MU population and showed no difference in NMJ transmission stability.

Therefore, could this technical difference reasonably explain how the NMJ molecular alterations were observed in the absence of DQEMG NMJ transmission stability changes? And taken a step further, suggest different adaptive responses in low- vs. mixed-threshold MU populations?

We thank the Reviewers for highlighting this point. We amended our abstract and key points accordingly.

2) Major conclusions/physiological mechanisms in humans.

I note the authors response to the first reviewer's comment number 21, subsection 1.

In human models of chronic disuse, such as ageing, there remains evidence that NMJs are stable across lifespan (Jones et al., 2017, Cell Reports 21:2348-2356). Likewise there is indirect support from EMG methods, like those used in the current manuscript, that the low-threshold, slow-type MU population has NMJ transmission resilience from chronic disuse in ageing (see reviews: Hepple and Rice, 2016, Journal of Physiology 594:1965-1978; Allen et al., 2021, Experimental Gerontology, 152:111465) and motor axon injury (example, Krarup et al., 2016, Clinical Neurophysiology 127:1675-1682). From this, we would expect that the low-threshold MU would be more likely to survive insult and re-innervate muscle fibres that become denervated.

With this view, without changes of MUP area or NF MUP jitter and jiggle at LS10 (acute disuse) does this provide support for NMJ functional (i.e., NMJ transmission stability) resilience? Can this NMJ functional resilience to acute disuse be generalized across both the low- and high-threshold MU populations as done in the abstract and key findings?

We thank the Reviewer for these important considerations. Although ageing is well known to be accompanied by reduced physical activity levels (McPhee *et al.*, 2016), we believe that ageing and disuse represent two well distinct phenomena. Ageing and disuse certainly share similar effects on skeletal muscle (e.g. reduced muscle mass and function) but also may induce differential adaptations. For instance, ageing is known to promote a shift towards the slow phenotype (i.e. increased type I fibre percentage), while disuse to a fast phenotype (i.e. increased type II fibre percentage) (Ciciliot *et al.*, 2013). Therefore, we believe that caution is needed when translating findings of ageing literature to disuse scenarios. Moreover, the literature on NMJ alterations with aging is large and complex. Some studies reported no changes in NMJ structure as Jones et al. (Jones *et al.*, 2017), but the general view is that ageing affects NMJs (Willadt *et al.*, 2018; Deschenes *et al.*, 2022). Also the two reviews cited by the Reviewer collectively support the concept that NMJ is altered with ageing, including also NMJ function at low contraction intensity (Hepple & Rice, 2016; Allen *et al.*, 2021):

However, we agree with the Reviewer that one limitation of studying NMJ transmission in vivo is that of mainly testing the low-threshold MUs and not the high-ones, as explained now in the abstract, key points, discussion and conclusions of the manuscript, and this is due to technical/methodological constraints. Nevertheless, considering that force generation proceeds through the Henneman size principle, we believe that even in maximal contraction, there is a contribution of the low-threshold motor units to the total force development. In fact, glycogen depletion experiments dating back to the '70s (Gollnick *et al.*, 1974) show that a switch from slow twitch to fast fibres twitch occurs at levels of about 20% of maximal exercise intensity in vastus lateralis.

Referee #2:

The authors have replied and amended many of my suggested changes. At places where they have not made changes mostly comes down to personal preferences/beliefs (fiber diameter or CSA) and whether a specific type of statistics is suitable for the data (EMG data). I do not have enough insight to judge the latter, but this comment does not seem reasonable to me: "i) considers the whole population of sampled decomposed motor units and not just the mean values obtained from each participant"

Three points that are still lacking:

The biopsy sampling sites seem very closely located. This is important information to provide to the readers. 2-3 cm normally refers to longitudinal spacing, and from the image it does not seem to be 2-3 cm between sites. Sampling from prior biopsy sites is associated with altered gene expression.

We thank the Reviewer for the observation. The image is just a representative graphical illustration of the biopsy sampling locations and could be slightly inaccurate. We maintained a distance of ~2/3 cm between biopsy sites that we believe is sufficient to avoid the effects of pre-sampling, as stated in the manuscript (section 2.7). It should also be considered that a greater longitudinal distance between the three biopsy sites could have influenced our immunohistochemistry and ATPase results since they are dependent on muscle region in which the biopsy is performed (e.g. Lexell *et al.*, 1983; Horwath *et al.*, 2021).

The authors did not follow up on my recommendation to do a quick a-BTX staining (very simple, easily done), which could have increased confidence in the RNAseq data not being confounded by NMJs.

We thank the Reviewer for this suggestion. As we explained in R1, we performed a-BTX staining in an additional biopsy part (~30/40mg) fixed in paraformaldehyde that was originally dedicated to NMJ morphology analysis. NMJs were detected in 29% of the samples (4/11 at LS0, 1/11 at LS10 and 4/9 at AR21). Representative images of a subject in which NMJs were found are provided below (Figure A). Unfortunately, we believe that the exact number of NMJs in these samples is unlikely representative of the number of NMJs in the biopsy part dedicated to RNAseq. Indeed, the biopsy portion used for RNAseq was smaller (~10/15mg) and, despite being part of the same biopsy, we cannot be sure that the two pieces had the same NMJs density. The same considerations can be extended to the biopsy part dedicated to immunohistochemistry analysis and it should also be considered that the NMJs presence change along the fibres longitudinal axis (Figure A). For these reasons, it seems hard to obtain a representative number from transversal sections.

Top

Fibre longitudinal axis

Bottom

Figure A: Representative images from a subject of our study in which NMJs were found. Location of NMJ is very heterogeneous along fibre length. Images are presented as Bright field to display myofibres (left panels) and immunofluorescence using a BTX-555 to detect acetylcholine receptors (AChR) (central panels).

Lastly, the authors refrain from showing the NCAM data in the manuscript, which is problematic, as they refer to them in the results and abstract: "and denervation status were assessed from blood samples and VL biopsies.". They should either remove completely or show data.

We added the NCAM data in Figure 8, as suggested.

References

- Allen MD, Dalton BH, Gilmore KJ, Mcneil CJ, Doherty TJ, Rice CL & Power GA (2021). Neuroprotective effects of exercise on the aging human neuromuscular system. *Exp Gerontol*; DOI: 10.1016/j.exger.2021.111465.
- Ciciliot S, Rossi AC, Dyar KA, Blaauw B & Schiaffino S (2013). Muscle type and fiber type specificity in muscle wasting. *Int J Biochem Cell Biol* **45**, 2191–2199.
- Deschenes MR, Flannery R, Hawbaker A, Patek L & Mifsud M (2022). Adaptive Remodeling of the Neuromuscular Junction with Aging.
- Gollnick P, Piehl K & Saltin B (1974). Selective glycogen depletion pattern in human muscle fibres after exercise of varying intensity and at varying pedalling rates. *J Physiol* **241**, 45–57.
- Hepple RT & Rice CL (2016). Innervation and neuromuscular control in ageing skeletal muscle. *J Physiol* **594**, 1965–1978.
- Horwath O, Envall H, Roja J, Emanuelsson EB, Sanz G, Ekblom B, Apro W & Moberg M (2021). Variability in vastus lateralis fiber type distribution, fiber size, and myonuclear content along and between the legs. *J Appl Physiol* **131**, 158–173.
- Jones RA, Harrison C, Eaton SL, Llaverro Hurtado M, Graham LC, Alkhamash L, Oladiran OA, Gale A, Lamont DJ, Simpson H, Simmen MW, Soeller C, Wishart TM & Gillingwater TH (2017). Cellular and Molecular Anatomy of the Human Neuromuscular Junction. *Cell Rep* **21**, 2348–2356.
- Lexell J, Henriksson-Larsen K & Sjostrom M (1983). Distribution of different fibre types in human skeletal muscles 2. A study of cross-sections of whole m. vastus lateralis. *Acta Physiol Scand* **117**, 115–122.
- McPhee JS, French DP, Jackson D, Nazroo J, Pendleton N & Degens H (2016). Physical activity in older age: perspectives for healthy ageing and frailty. *Biogerontology* **17**, 567–580.
- Willadt S, Nash M & Slater C (2018). Age-related changes in the structure and function of mammalian neuromuscular junctions. *Ann N Y Acad Sci* **1412**, 41–53.

Dear Dr Sarto,

Re: JP-RP-2022-283381R2 "Effects of short-term unloading and active recovery on human motor unit properties, neuromuscular junction transmission and transcriptomic profile" by Fabio Sarto, Dan Stashuk, Martino V Franchi, Elena Monti, Sandra Zampieri, Giacomo Valli, Giuseppe Sirago, Julian Candia, Lisa M Hartnell, Matteo Paganini, Jamie S McPhee, Giuseppe De Vito, Luigi Ferrucci, Carlo Reggiani, and Marco Narici

I am pleased to tell you that your paper has been accepted for publication in The Journal of Physiology.

NEW POLICY: In order to improve the transparency of its peer review process The Journal of Physiology publishes online as supporting information the peer review history of all articles accepted for publication. Readers will have access to decision letters, including all Editors' comments and referee reports, for each version of the manuscript and any author responses to peer review comments. Referees can decide whether or not they wish to be named on the peer review history document.

The last Word version of the paper submitted will be used by the Production Editors to prepare your proof. When this is ready you will receive an email containing a link to Wiley's Online Proofing System. The proof should be checked and corrected as quickly as possible.

Authors should note that it is too late at this point to offer corrections prior to proofing. The accepted version will be published online, ahead of the copy edited and typeset version being made available. Major corrections at proof stage, such as changes to figures, will be referred to the Reviewing Editor for approval before they can be incorporated. Only minor changes, such as to style and consistency, should be made a proof stage. Changes that need to be made after proof stage will usually require a formal correction notice.

All queries at proof stage should be sent to TJP@wiley.com

Are you on Twitter? Once your paper is online, why not share your achievement with your followers. Please tag The Journal (@jphysiol) in any tweets and we will share your accepted paper with our 23,000+ followers!

Yours sincerely,

Scott K. Powers
Senior Editor
The Journal of Physiology
<https://jp.msubmit.net>
<http://jp.physoc.org>
The Physiological Society
Hodgkin Huxley House
30 Farringdon Lane
London, EC1R 3AW
UK
<http://www.physoc.org>
<http://journals.physoc.org>

P.S. - You can help your research get the attention it deserves! Check out Wiley's free Promotion Guide for best-practice recommendations for promoting your work at www.wileyauthors.com/eoo/guide. And learn more about Wiley Editing Services which offers professional video, design, and writing services to create shareable video abstracts, infographics, conference posters, lay summaries, and research news stories for your research at www.wileyauthors.com/eoo/promotion.

* IMPORTANT NOTICE ABOUT OPEN ACCESS *

To assist authors whose funding agencies mandate public access to published research findings sooner than 12 months after publication The Journal of Physiology allows authors to pay an open access (OA) fee to have their papers made freely available immediately on publication.

You will receive an email from Wiley with details on how to register or log-in to Wiley Authors Services where you will be able to place an OnlineOpen order.

You can check if your funder or institution has a Wiley Open Access Account here <https://authorservices.wiley.com/author-resources/Journal-Authors/licensing-and-open-access/open-access/author-compliance-tool.html>

Your article will be made Open Access upon publication, or as soon as payment is received.

If you wish to put your paper on an OA website such as PMC or UKPMC or your institutional repository within 12 months of publication you must pay the open access fee, which covers the cost of publication.

OnlineOpen articles are deposited in PubMed Central (PMC) and PMC mirror sites. Authors of OnlineOpen articles are permitted to post the final, published PDF of their article on a website, institutional repository, or other free public server, immediately on publication.

Note to NIH-funded authors: The Journal of Physiology is published on PMC 12 months after publication, NIH-funded authors DO NOT NEED to pay to publish and DO NOT NEED to post their accepted papers on PMC.

EDITOR COMMENTS

Reviewing Editor:

The reviewers are satisfied with the revised version. Congratulations for an excellent study, with potential impact in the field.

Senior Editor:

Thank you for submitting your excellent report to the Journal of Physiology.

REFEREE COMMENTS

Referee #1:

The authors have made improvements that has resulted in clearer communication of their findings. I'm fully supportive of this manuscript, well done.

Referee #2:

I appreciate the authors response. I agree that 2-3 cm between sampling sites is reasonable (disregarding what the image in revision 1 showed). I have nothing further to add at this point.

2nd Confidential Review

24-Aug-2022